# Evaluation of the Growth Assessment Protocol (GAP) for antenatal detection of small for gestational age: The DESiGN cluster randomised trial

Matias C. Vieira[1,2], Sophie Relph[1], Walter Muruet-Gutierrez[1,3], Maria Elstad[3], Bolaji Coker[3,4], Natalie Moitt[1], Louisa Delaney[1], Chivon Winsloe[1,5], Andrew Healey[6], Kirstie Coxon[7], Alessandro Alagna[8], Annette Briley[1,9], Mark Johnson[10], Louise M. Page[11], Donald Peebles[12], Andrew Shennan[1], Baskaran Thilaganathan[13,14], Neil Marlow[12], Lesley McCowan[15], Christoph Lees[10], Deborah A. Lawlor[16,17,18], Asma Khalil[13,14], Jane Sandall[1], Andrew Copas[5], Dharmintra Pasupathy[1,19]*, on behalf of the DESiGN Collaborative Group[¶]

1 Department of Women and Children's Health, King's College London, London, United Kingdom, 2 Department of Obstetrics and Gynaecology, University of Campinas (UNICAMP), Campinas, Brazil, 3 School of Population Health and Environmental Sciences, King's College London, London, United Kingdom, 4 NIHR Biomedical Research Centre at Guy's and St Thomas' NHS Foundation Trust and King's College London, London, United Kingdom, 5 Centre for Pragmatic Global Health Trials, University College London, London, United Kingdom, 6 Centre for Implementation Science and King's Health Economics, King's College London, London, United Kingdom, 7 Faculty of Health, Social Care and Education, Kingston University and St. George's, University of London, London, United Kingdom, 8 London Perinatal Morbidity and Mortality Working Group (NHS), London, United Kingdom, 9 Caring Futures Institute Flinders University and North Adelaide Local Health Network, Adelaide, Australia, 10 Department of Metabolism, Digestion and Reproduction, Imperial College London, London, United Kingdom, 11 West Middlesex University Hospital, Chelsea & Westminster Hospital NHS Foundation Trust, Isleworth, United Kingdom, 12 UCL Institute for Women's Health, University College London, London, United Kingdom, 13 Fetal Medicine Unit, St George's University Hospitals NHS Foundation Trust, London, United Kingdom, 14 Molecular & Clinical Sciences Research Institute, St George's, University of London, London, United Kingdom, 15 Faculty of Medical and Health Sciences, University of Auckland, Auckland, New Zealand, 16 Bristol NIHR Biomedical Research Centre, Bristol, United Kingdom, 17 Medical Research Council Integrative Epidemiology Unit at the University of Bristol, Bristol, United Kingdom, 18 Population Health Science, University of Bristol, Bristol, United Kingdom, 19 Reproduction and Perinatal Centre, University of Sydney, Sydney, Australia

¶ Membership of DESiGN Collaborative Group is provided in the Acknowledgments.
* Dharmintra.Pasupathy@sydney.edu.au

**Data Availability Statement:** Data cannot be shared publicly because consent was not obtained from women; permission for sharing data was not

## Abstract

### Background

Antenatal detection and management of small for gestational age (SGA) is a strategy to reduce stillbirth. Large observational studies provide conflicting results on the effect of the Growth Assessment Protocol (GAP) in relation to detection of SGA and reduction of still-birth; to the best of our knowledge, there are no reported randomised control trials. Our aim was to determine if GAP improves antenatal detection of SGA compared to standard care.

### Methods and findings

This was a pragmatic, superiority, 2-arm, parallel group, open, cluster randomised control trial. Maternity units in England were eligible to participate in the study, except if they had

sought as part of ethical approval. Data is only available following approval from Research Ethics Committee and Confidentiality Advisory Group. Enquiries and requests should be made to DESiGN trial team and sponsors through the Department of Women and Children's Health at King's College London (SoLCS_research@kcl.ac.uk).

**Funding:** This study was funded by Guy's and St Thomas' Charity (MAJ150704), Stillbirth and Neonatal Death Charity - SANDS (RG1011/16) and Tommy's Charity. MCV was supported by CAPES (BEX 9571/13–2). SR, KC, AH and JS were supported by the National Institute for Health Research (NIHR) Collaboration for Leadership in Applied Health Research and Care South London at King's College Hospital NHS Foundation Trust. NM receives a proportion of funding from the Department of Health's NIHR Biomedical Research Centres funding scheme at UCLH/UCL. DAL's contributions were supported by the Bristol NIHR Biomedical Research Centre and her NIHR Senior Investigator Award (NF-0616-10102). JS is supported by an NIHR Senior Investigator Award. DP was funded by Tommy's Charity during the period of the study. The funders had no role in study design, data collection and analysis, decision to publish, or preparation of the manuscript.

**Competing interests:** I have read the journal's policy and the authors of this manuscript have the following competing interests: NM reports personal fees from Takeda, personal fees from RSM Consulting, personal fees from Novartis, outside the submitted work. BT is the Clinical Director of the Tommy's National Centre for Maternity Improvement based at the Royal College of Obstetrics and Gynaecology (RCOG); the Centre's objective is to translate the latest evidence into clinical practice in the UK. DAL has received support from Medtronic Ltd and Roche Diagnostics for research unrelated to that presented here. LP is clinical advisor [and from Sept 2021 deputy clinical director] for Healthcare Safety Investigation Branch maternity investigation programme, President of the British Intrapartum Care Society (BICS), invited member of some RCOG working groups and co-opted member of the British Maternal and Fetal Medicine Society (BMFMS) committee; she also received support from Pharmacosmos for clinical consultancy in work unrelated to that presented here.

**Abbreviations:** CI, confidence interval; DESiGN, DEtection of Small for GestatioNal age fetus; EFW, estimated fetal weight; GAP, Growth Assessment Protocol; GROW, Gestation-Related Optimal Weight; HRA, Health Research Authority; mITT,

already implemented GAP. All women who gave birth in participating clusters (maternity units) during the year prior to randomisation and during the trial (November 2016 to February 2019) were included. Multiple pregnancies, fetal abnormalities or births before $24^{+1}$ weeks were excluded. Clusters were randomised to immediate implementation of GAP, an antenatal care package aimed at improving detection of SGA as a means to reduce the rate of stillbirth, or to standard care. Randomisation by random permutation was stratified by time of study inclusion and cluster size. Data were obtained from hospital electronic records for 12 months prerandomisation, the washout period (interval between randomisation and data collection of outcomes), and the outcome period (last 6 months of the study). The primary outcome was ultrasound detection of SGA (estimated fetal weight <10th centile using customised centiles (intervention) or Hadlock centiles (standard care)) confirmed at birth (birthweight <10th centile by both customised and population centiles). Secondary outcomes were maternal and neonatal outcomes, including induction of labour, gestational age at delivery, mode of birth, neonatal morbidity, and stillbirth/perinatal mortality. A 2-stage cluster–summary statistical approach calculated the absolute difference (intervention minus standard care arm) adjusted using the prerandomisation estimate, maternal age, ethnicity, parity, and randomisation strata. Intervention arm clusters that made no attempt to implement GAP were excluded in modified intention to treat (mITT) analysis; full ITT was also reported. Process evaluation assessed implementation fidelity, reach, dose, acceptability, and feasibility. Seven clusters were randomised to GAP and 6 to standard care. Following exclusions, there were 11,096 births exposed to the intervention (5 clusters) and 13,810 exposed to standard care (6 clusters) during the outcome period (mITT analysis). Age, height, and weight were broadly similar between arms, but there were fewer women: of white ethnicity (56.2% versus 62.7%), and in the least deprived quintile of the Index of Multiple Deprivation (7.5% versus 16.5%) in the intervention arm during the outcome period. Antenatal detection of SGA was 25.9% in the intervention and 27.7% in the standard care arm (adjusted difference 2.2%, 95% confidence interval (CI) −6.4% to 10.7%; $p = 0.62$). Findings were consistent in full ITT analysis. Fidelity and dose of GAP implementation were variable, while a high proportion (88.7%) of women were reached. Use of routinely collected data is both a strength (cost-efficient) and a limitation (occurrence of missing data); the modest number of clusters limits our ability to study small effect sizes.

## Conclusions

In this study, we observed no effect of GAP on antenatal detection of SGA compared to standard care. Given variable implementation observed, future studies should incorporate standardised implementation outcomes such as those reported here to determine generalisability of our findings.

## Trial registration

This trial is registered with the ISRCTN registry, ISRCTN67698474.

modified intention to treat; SGA, small for
gestational age; WHO, World Health Organization.

## Why was this study done?

- Antenatal detection and appropriate management of small for gestational age (SGA) infants is a recognised strategy to prevent stillbirth; previous reports have suggested the rate of stillbirth is halved when SGA is antenatally detected, compared to undetected SGA.

- Large observational studies provide conflicting results on the effect of Growth Assessment Protocol (GAP), an antenatal care package, with both findings of increased and no difference in detection of SGA and reduction of stillbirth.

- The observational nature of all previous studies about GAP limits the assessment of causality in any observed associations.

## What did the researchers do and find?

- To the best of our knowledge, this is the first randomised control trial of GAP, comparing 11,096 births exposed to the intervention (5 clusters) to 13,810 exposed to standard care (6 clusters) during the outcome period.

- We observed no significant effect on antenatal detection of SGA compared to standard care (25.9% versus 27.7%; adjusted difference 2.2%, 95% confidence interval (CI) −6.4% to 10.7%).

- The lack of effect should be interpreted in the context of the variable implementation of GAP.

## What do these findings mean?

- This randomised control trial of GAP compared to standard care did not observe improvement in ultrasound detection of SGA; variable implementation of GAP was observed consistent with previous studies.

- It is imperative that future studies of GAP assess implementation using standardised outcomes (fidelity, reach, and dose), in order to determine generalisability of our findings, identify barriers to implementation, and hence better inform policy for improving perinatal outcomes.

- Use of routinely collected data is both a strength (cost-efficient) and a limitation (occurrence of missing data); the modest number of hospitals in this study limits our ability to study small differences between groups.

## Introduction

In 2014, the World Health Organization (WHO) launched the Every Newborn Action Plan with the aim to end preventable perinatal deaths by 2030; reducing stillbirth is thus a global

priority [1]. While national strategies to tackle stillbirth vary according to leading causes locally, the importance of risk stratification and screening strategies that target improved detection of small for gestational age (SGA) (birthweight <10th centile) and appropriate management and timely delivery has been emphasised for high-income countries [2,3]. Antenatal detection of SGA has been associated with a halved risk of stillbirth compared to undetected SGA [4,5]. A review of guidelines from 6 high-income countries described a consensus on recommendations for stratifying women by risk of SGA, but noted variation in other aspects of screening and management, such as the use of customised fetal charts to identify SGA and the role of universal third trimester ultrasound [6].

The Growth Assessment Protocol (GAP), developed by the Perinatal Institute [7], is a complex intervention that includes the use of customised centile charts for fundal height and estimated fetal weight (EFW) measurements (Gestation-Related Optimal Weight (GROW) charts), evidence-based protocols and risk assessment, training and accreditation of clinical staff, a rolling audit programme and benchmarking of performance [8]. A nonrandomised control trial in the United Kingdom (UK) of standardised fundal height measurements plotted on customised charts demonstrated an increase in antenatal detection of SGA (29% versus 48%, odds ratio 2.2; 95% confidence interval (CI) 1.1 to 4.5) [9]. A recent study in New Zealand reported an almost 3-fold increase in detection of SGA (22.9% versus 57.9%; $p < 0.001$) when comparing rates before and after implementation of GAP [10]. In the UK, national uptake of GROW charts or GAP increased between 2007 and 2012 with a concomitant 22% reduction of stillbirth rates in regions of high uptake [11]. However, a study comparing the trend of stillbirth rates during 2010 to 2015 in England and Wales to that in Scotland where uptake of GAP was very low reported a greater decline in Scotland [12]. The authors concluded that any association between GAP and reductions in stillbirth rates was coincidental rather than causal. To our knowledge, there has been no randomised control trial studying the impact of GAP versus standard care on detection of SGA. There is also paucity of data on the impact of GAP on service usage (e.g., number of ultrasound scans and induction of labour) and on unwanted potential effects, such as a possible increase in neonatal adverse outcomes related to iatrogenic late preterm/early term birth.

The primary aim of the DESiGN trial (DEtection of Small for GestatioNal age fetus) was to determine whether implementation of GAP results in improved ultrasound detection of SGA, when compared to standard care. We also planned to explore the effect on related maternal and neonatal outcomes and to conduct a process evaluation of fidelity, reach, dose, acceptability, feasibility, and resource use.

## Methods

### Study design and population

The DESiGN trial was a pragmatic, superiority, 2-arm, parallel group, open, cluster randomised control trial, including 13 maternity units in England [13]. All women who gave birth in participating clusters (maternity units) during the trial (between November 2016 and February 2019) were included. Baseline data were also collected on women who gave birth during the year prior to cluster randomisation. Pregnancies with significant fetal abnormalities, multiple pregnancies, and pregnancies ending before $24^{+1}$ weeks of gestation (referred to as weeks in the paper) were excluded. The study design and methodology of this trial have been prospectively registered (ISRCTN67698474), and both the study protocol (S1 Protocol) and the pre-specified analysis plan (S1 Appendix) have been approved by the Trial Steering Committee.

We enrolled maternity units primarily in London given the lower uptake of GAP in this area at the time the trial was proposed compared to the whole of the UK, where uptake was

64% [14]. A cluster trial was undertaken because the intervention requires implementation of site-wide guidelines for screening and management of SGA and additional staff training. Within-site contamination would limit the validity of individual randomisation. The trial was pragmatic to capture the reality of the introduction of this complex intervention into clinical practice with support from the Perinatal Institute.

## Randomisation and masking

Clusters were randomly allocated by the trial statistician to immediate implementation of GAP (intervention arm) or to continue standard care during the study period (standard care arm). Randomisation occurred in 3 strata according to time of inclusion in the study (8, 3, and 2 clusters, respectively); the randomisation of the first 8 clusters were further stratified by size of maternity unit (number of births during the year 2013 to 2014). Randomisation was by random permutation within strata, providing exact 1:1 allocation except in the second stratum of 3 clusters where it was determined at random which arm would receive 2 clusters. The random permutation was conducted in Stata v14 (StataCorp LP, College Station, Texas, USA). Due to the nature of the intervention, concealment was not possible.

## Procedures

Data were collected from a prerandomisation period of 12 consecutive months, which differed by randomisation stratum, the washout period (variable duration) during which the intervention arm clusters were implementing GAP, and for an outcome comparison period (outcome period) of 4 to 6 months from 1 September 2018 to 28 February 2019. The outcome period commenced when women giving birth in intervention clusters had had time to receive full antenatal exposure to GAP. One cluster from the control arm provided outcome data earlier due to a previously planned introduction of GAP at the original trial end date. This was a consequence of the washout period being extended after delays in GAP implementation at the last cluster randomised to the intervention.

Data were obtained from 4 types of routinely collected electronic patient record system at each cluster: maternity, ultrasound, neonatal, and administrative [15]. Additional data were collected to assess compliance with the intervention in allocated clusters from review of a subset of women's paper maternity records ($n$ = 120 per cluster). Data were anonymised locally by the trial team before being sent centrally for data management, storage, and analysis.

Following randomisation, maternity units allocated to the intervention were expected to contact the providers of GAP to commence training and implementation support. The components of GAP implementation are detailed in Table 1, by stage of implementation. Following consultation with cluster sites, the e-learning training requirement was amended by the Perinatal Institute to allow compliance with e-learning certification to be achieved within 3 months of going "live." The prespecified requirements that describe how an implementing cluster would be considered as GAP compliant are further detailed in the study protocol (S1 Protocol; page 74). These were GAP recommendations during this trial; there were changes introduced subsequent to this study [16].

In the standard care arm, women received routine antenatal care as per the local guidelines for screening and management of SGA in each cluster. There was no prespecification of policies in this arm, except that these clusters should not implement GAP or use customised centiles for fundal height or ultrasound monitoring of fetal growth. At the time this trial started, standard care for screening and management of SGA was guided by an RCOG guideline [17]. This recommends stratification of pregnant women by presence of risk factors for SGA. Women at low risk of SGA are further screened using measurement of fundal height at each

**Table 1. Expected components of GAP implementation.**

| Implementation Stage | GAP requirements |
|---|---|
| Preparation and planning | • Nominated staff from each cluster to attend "Train the Trainers" GAP workshop.<br>• Cluster to conduct a baseline audit of SGA detection (10% of annual births).<br>• Cluster to prepare local guideline for the "Assessment of Fetal Growth" modelled on GAP recommendations. |
| Implementation | • Cluster trainers to cascade face-to-face training to 75% of colleagues from each professional group (midwives, obstetricians, sonographers).<br>• GAP e-learning module to also be completed by 75% staff members from each professional group. |
| Ongoing use of GAP | • Access to GROW chart online programme provided by the Perinatal Institute after cluster compliant with above requirements.<br>• Each pregnant woman assessed for risk of SGA at antenatal booking appointment using GAP tool.<br>• Customised GROW chart printed for each pregnant woman at antenatal booking appointment and used to assess fetal growth by plotting fundal height measurements or estimated fetal weight on the chart.<br>• Women at low risk of SGA expected to have a fundal height measured 3-weekly during pregnancy, commencing between 26 and 28 weeks. If plots deviate from what is expected (first plot below 10th centile, slow/static/accelerative growth), the woman should be referred for a fetal growth scan.<br>• Women at high risk of SGA expected to have an ultrasound scan to estimate fetal weight 3-weekly during pregnancy, commencing between 26 and 28 weeks.<br>• Where GROW chart EFW plots deviate from the expected trajectory (as per fundal height deviations), RCOG protocols should be followed for further investigation of suspected SGA [17].<br>• Birthweight centiles are calculated at the time of birth using the GROW software. This also prompts the clinician to enter whether SGA was detected antenatally, to inform auditing of practice and national benchmarking.<br>• GAP users are encouraged to use the GAP online proforma to conduct analyses of 'missed cases' in which SGA was not detected antenatally. |

EFW, estimated fetal weight by ultrasound; GAP, Growth Assessment Protocol; GROW, Gestation-Related Optimal Weight chart; RCOG, Royal College of Obstetricians and Gynaecologists; SGA, small for gestational age.

antenatal appointment after 24 weeks. Women with risk factors are either offered serial fetal growth ultrasound scans or further stratification using doppler assessment of the uterine arteries at 20 weeks of gestation, dependent on the number or significance of the risk factors present. RCOG does not guide frequency of serial growth scans. Following a request from reviewers, a summary description of recommended practice in standard care clusters is provided on S2 Appendix (page 2) based on review of local guidelines for screening and detection of SGA. The Saving Babies' Lives care bundle is a complex antenatal intervention that started to be implemented nationally during the trial. Clusters in the standard care arm were exempted from compliance with element 2 (risk assessment and surveillance of fetal growth restriction) of the Saving Babies' Lives bundle. However, it was considered unethical to stop clusters in the standard care arm that were willing to implement concomitant strategies for improved detection of SGA and prevention of stillbirths initiated locally or nationally, which could include the Saving Babies' Lives care bundle [18].

## Process evaluation of implementation

The process evaluation examined implementation compliance, acceptability, feasibility, contextual factors, and mechanisms of impact. To assess compliance with the intervention in implementing sites, we assessed fidelity, reach, and dose [19], by comparing site guidelines to those recommended by GAP, assessing compliance with training targets and by a review of 600 women's maternity records (40 randomly selected singleton nonanomalous births in each of 3 months during the outcome period at 5 implementing clusters). Acceptability and feasibility of GAP implementation were explored through interviews with clinicians including clinical

leads. A summary of implementation is provided in this report to support interpretation of the main findings (methodology provided in S2 Appendix; page 3). We also collected guideline on screening for SGA from clusters in the standard care arm. A more detailed process evaluation analysis will be reported separately.

## Outcomes

The primary outcome of this study was antenatal ultrasound detection of SGA (after 24 completed weeks) defined for infants who are SGA (i.e., birthweight less than 10th centile) according to both population (UK1990 birthweight centiles) and customised (GROW) charts [20,21]. This definition was chosen because GAP targets detection of babies who are SGA by customised centiles, whereas standard care largely uses population centile charts. Antenatal detection of SGA was defined as ultrasound-derived EFW <10th centile by customised (GROW) charts in the intervention arm during the outcome period and by population [22] fetal charts for babies born in intervention sites during the prerandomisation period and all babies born in the standard care arm [20–22]. For calculation of ultrasound detection of SGA, data were obtained from electronic ultrasound records to identify EFW <10th centile and from electronic maternity records to identify birthweight <10th centile; these were calculated for all births in each cluster. A detailed description of methodology for calculating the rate of antenatal detection of SGA is provided in S2 Appendix (page 4).

The 26 planned secondary outcomes included the test positive rate for antenatal detection of SGA (defined by both definitions as per primary outcome), antenatal detection and false positive rate of antenatal ultrasound detection of SGA confirmed at birth by customised centiles and by population centiles, maternal outcomes (induction of labour, mode of birth, postpartum haemorrhage, severe perineal tear (third/fourth degree), epidural and episiotomy), neonatal parameters and measures of condition at birth (gestational age at birth, preterm birth, birthweight, Apgar score <7 at 5 minutes, arterial cord pH <7.1, respiratory support at birth), neonatal unit admission, major neonatal morbidity (defined as one or more of: receipt of supplemental oxygen at 28 days of age, necrotising enterocolitis, sepsis, neonatal retinopathy, hypoxic–ischemic encephalopathy, intraventricular haemorrhage), minor neonatal morbidity (defined as one or more of hypothermia, hypoglycaemia, nasogastric tube feeding), stillbirth, neonatal death, and perinatal mortality. Utilisation of ultrasound scan was a process outcome (proportion of pregnancies with a scan, proportion of pregnancies with a scan between $18^{+0}$ and $24^{+0}$ weeks, proportion of pregnancies with a scan after $24^{+0}$ weeks with EFW, number of scans per pregnancy after $24^{+0}$ weeks with EFW, proportion of pregnancies with no record of ultrasound). Timing of scans after 24 weeks (i.e., utilisation per week gestation) was described following a request from reviewers and the academic editor, with the aim of better understanding differences in practice between trial arms. These process measures were reported to provide context to results.

## Statistical analysis

Data management was performed to harmonise and amalgamate datasets from all clusters. This process has previously been described in detail and published [15]. The approach for multiple imputation of missing data is summarised in the S2 Appendix (page 5).

Characteristics of the individual participants in the prerandomisation and trial outcome period were summarised for each trial arm using means and standard deviations, medians and interquartile ranges or frequencies and percentages, as appropriate. These results are reported using imputed data, where available; results from available case analyses are provided in the Supporting information.

**Main analyses.** The primary analysis was performed using a modified intention to treat (mITT) approach. This involved excluding any cluster in the intervention arm that did not contact the GAP provider to initiate implementation of the intervention due to changes in local strategy, since such changes are not considered informative of how GAP would have performed in the cluster. Due to the modest number of clusters, the analysis was performed using an unweighted 2-stage cluster-summary statistical approach [23]; detailed description provided in S2 Appendix (page 6). Intervention effects (absolute difference of intervention minus standard care arm) are presented with 95% CIs. A sensitivity analysis was also performed at the request of reviewers, excluding 1 cluster without ultrasound measurement data for the baseline period, which are imputed in our main analysis (S2 Appendix; page 5).

**Prespecified secondary, subgroup, and sensitivity analyses.** A secondary analysis was planned using a per protocol approach restricting analysis of the intervention arm to clusters that complied with the GAP preimplementation requirements (S1 Protocol; page 74) in full. A further secondary analysis was a full intention to treat (ITT) analysis in which data from all clusters were used as randomised, irrespective of whether or not GAP was implemented. A prespecified subgroup analysis was planned to explore the effect of the intervention on 21 clinical and neonatal outcomes, only in SGA infants. A sensitivity analysis explored the intervention effect when restricted only to women who had an ultrasound scan between $18^{+0}$ and $24^{+0}$ weeks (presumed fetal anomaly scan) at the cluster where she later gave birth, reflecting antenatal care primarily within a single cluster and consistent exposure to the intervention from 24 weeks. A reviewer requested a further post hoc sensitivity analysis of the stillbirth outcome, concerned that our 2-stage analysis approach may be unsuitable for rare outcomes. After preferred 1-stage methods were found unfeasible or did not converge, we applied the standard logistic regression approach but with robust standard errors to acknowledge clustering (see S2 Appendix, page 6 for details). We use the standard 5% significance level for testing across our secondary outcomes and subgroup and sensitivity analyses. Due to multiple testing, significant results for secondary outcomes should be treated with caution.

These analyses were conducted following a prespecified analysis plan (S1 Appendix). All prespecified subgroup and sensitivity analyses were detailed in the trial protocol (S1 Protocol) and approved by the trial steering committee. This study has been reported as per the Consolidated Standards of Reporting Trials (CONSORT) statement (S1 CONSORT Checklist).

**Sample size calculation.** The power calculation for this study determined a minimum target sample size of 12 clusters (6 per arm) based on information collected during protocol development [13]. We were unable to identify reports of an intracluster correlation coefficient for detection of SGA; therefore, a coefficient of the most approximate outcome (rate of fetal growth restriction) was used (0.019) [24]. A cluster size that included an average of 126 SGA infants (defined by customised and population centile charts) with 6 clusters per arm provides 84% power to detect an improvement in the detection of SGA, assuming 20% are detected using standard care and 33% detected using GAP (doubling of odds ratio for detection) at the 5% significance level (2-sided test) [13]. We made no explicit allowance for the additional baseline data from each cluster, their inclusion is likely to increase power. Power calculations were performed using the user-written programme "clustersampsi" for Stata.

## Protocol changes

The trial protocol was amended during the study period for logistical and methodological reasons, including changes to data flow and storage, and following a change to the trial sponsor in

2017. A further change occurred prior to the randomisation of recruited clusters, whereby the definition of the primary outcome was refined. The registration of this change was delayed until after randomisation because of the change in study sponsor. Nevertheless, the amendment was approved before any women included in the primary analysis had given birth. These and other minor study amendments are recorded in the current version of the study protocol (S1 Protocol). All amendments were approved by the Research Ethics Committee and participating sites' Research and Development departments. Approval was also sought from the trial steering committee, Confidentiality Advisory Group and funders, where appropriate. During data management and analysis, the definition of major neonatal morbidity changed in relation to the study protocol, as the data was insufficiently detailed to determine Bell stage of necrotising enterocolitis, culture status in sepsis, and need for ophthalmic intervention related to retinopathy.

### Ethical approval

Ethical approval for this trial was obtained from the Health Research Authority (HRA) through the London Bloomsbury Research Ethics Committee (Ref. 15/LO/1632) and the Confidentiality Advisory Group (Ref. 15/CAG/0195). Individual informed consent was not obtained, but women could request to opt out from sharing their data. A key professional for each cluster provided written cluster consent prior to randomisation.

### Patient and public involvement

Patient groups and stakeholders (representing both PPI and professional groups) were involved from the conceptualisation of this study. Patient groups were provided with a summary for the study and procedures in lay terms and asked their opinion about key points including the relevance of the study and the use of data without individual informed consent given the cluster intervention/design. Their feedback was used to inform the final study protocol and ethical application. Stakeholders such as Stillbirth Clinical Study Group from RCOG, SANDS Charity, and Tommy's Charity were also involved in the conceptualisation of this study. We have a patient representative in our coinvestigator group who has provided their perspective throughout the study, including in interpretation and explanation of results to a lay audience.

## Results

Among the 16 sites that were invited to participate in the trial, 13 were willing and enrolled in the study (Fig 1). Seven clusters were allocated to the intervention and 6 to standard care. Two sites randomised to the intervention did not contact the GAP provider to initiate implementation. The median washout period was 17 months (range 11 to 18), this included a median 9 months (range 6 to 12 months) interval between antenatal booking of women (presumed to be at 12 weeks) with the opportunity of exposure to GAP until commencement of the outcome period. Among the 209,314 pregnancies during the study period in the 13 participating sites, 201,209 were included in the study. For the primary analysis (mITT), the outcome period included 13,810 pregnancies in the standard care arm (6 clusters) and 11,096 pregnancies in the intervention arm (5 clusters). No women asked for their data to be excluded from the study.

Demographic characteristics are provided in Table 2. In the prerandomisation period, age, height, and weight were broadly similar between trial arms, but there were fewer women: of white ethnicity (55.9% versus 62.8%), with obesity (15.7% versus 18.1%), and in the first (least

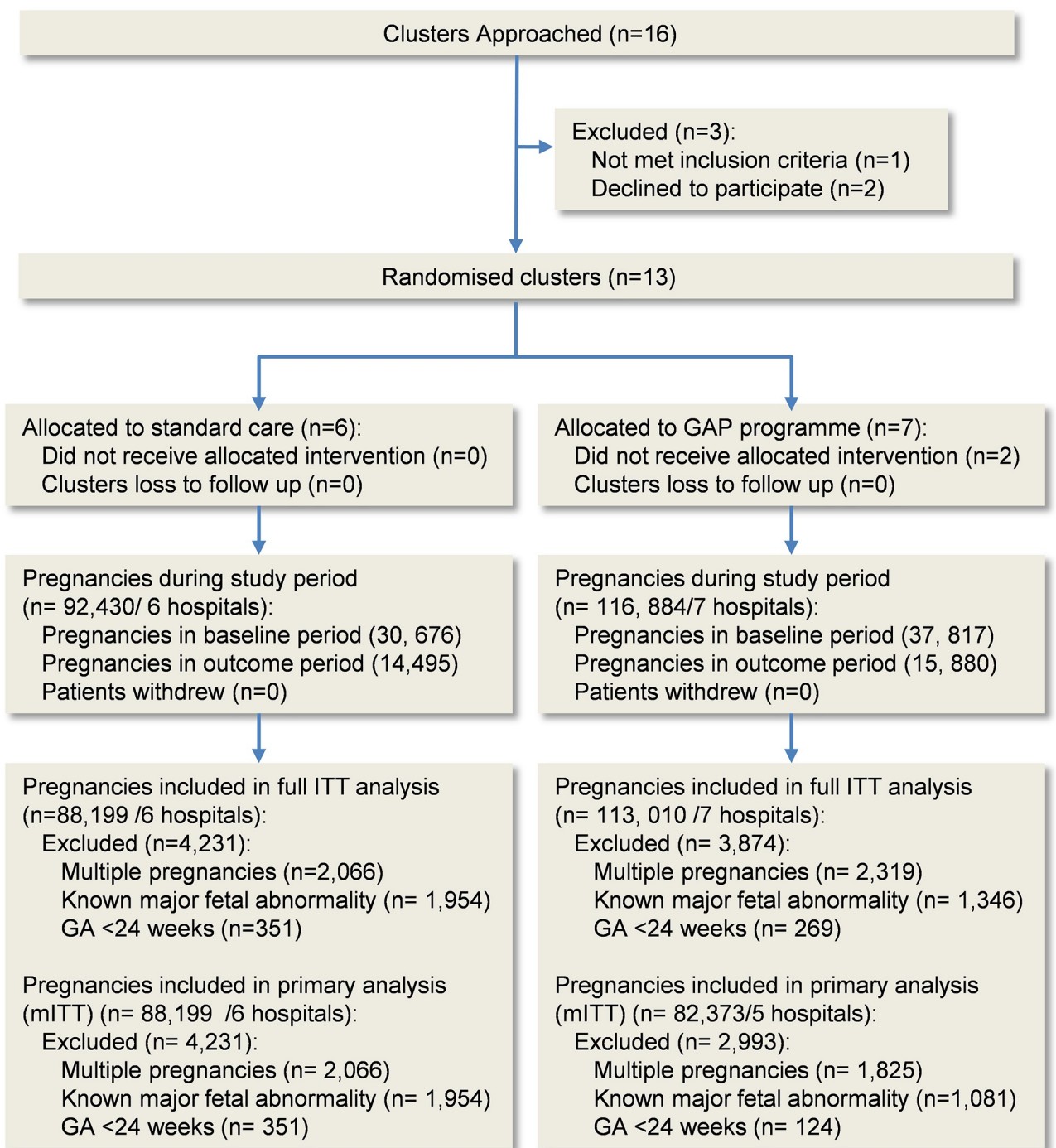

**Fig 1. Study population (CONSORT flow diagram).** GAP, Growth Assessment Protocol; ITT, intention to treat; mITT, modified intention to treat.

deprived) quintile of Index of Multiple Deprivation (7.6% versus 17.4%) in the intervention arm than the standard care arm. Similar findings were observed in the outcome period. Demographic characteristics were also broadly similar using available case data (for variables that were imputed) and the ITT sample (13 clusters) (Tables A and B in S3 Appendix). A

**Table 2. Clinical and sociodemographic characteristics according to treatment allocation (modified intention to treat analysis).**

| | Prerandomisation period | | Outcome period | |
|---|---|---|---|---|
| | Standard Care (*n* = 29,404) | Intervention (GAP) (*n* = 26,546) | Standard Care (*n* = 13,810) | Intervention (GAP) (*n* = 11,096) |
| **Imputed data** | | | | |
| Age at conception (years), median (IQR) | 31.6 (27.5, 35.2) | 31.5 (27.6, 35.2) | 32.0 (27.9, 35.4) | 31.8 (27.9, 35.5) |
| Ethnicity, % | | | | |
| White | 62.8 | 55.9 | 62.7 | 56.2 |
| Black | 16.2 | 12.7 | 15.1 | 12.6 |
| Asian | 13.3 | 19.4 | 13.5 | 20.3 |
| Mixed | 2.1 | 1.9 | 2.6 | 1.6 |
| Other | 5.5 | 10.1 | 6.1 | 9.2 |
| Index of Multiple Deprivation Quintiles, % | | | | |
| 1 (Least deprived) | 17.4 | 7.6 | 16.5 | 7.5 |
| 2 | 12.5 | 10.8 | 12.7 | 10.6 |
| 3 | 16.1 | 23.2 | 16.6 | 23.6 |
| 4 | 28.5 | 34.7 | 28.7 | 35.4 |
| 5 (Most deprived) | 25.4 | 23.7 | 25.5 | 22.9 |
| Maternal Height (m), median (IQR) | 1.64 (1.60, 1.69) | 1.64 (1.59, 1.68) | 1.64 (1.60, 1.69) | 1.64 (1.60, 1.68) |
| Maternal Weight (kg), median (IQR) | 66.0 (59.5, 76.0) | 65.6 (57.4, 74.0) | 67.0 (59.5, 77.9) | 65.4 (58.0, 76.0) |
| Body Mass Index Categories, % | | | | |
| <18.5 | 3.9 | 4.1 | 3.4 | 3.4 |
| (18.5–24.9) | 50.1 | 53.9 | 47.2 | 51.6 |
| (25.0–29.9) | 28.0 | 26.3 | 29.5 | 27.2 |
| (30.0–34.9) | 11.9 | 10.5 | 13.1 | 11.3 |
| (35.0–39.9) | 4.2 | 3.5 | 4.6 | 4.4 |
| ≥40.0 | 2.0 | 1.7 | 2.2 | 2.1 |
| Parity, % | | | | |
| Nulliparous | 46.4 | 59.0 | 47.5 | 51.6 |
| 1 | 33.8 | 26.3 | 34.0 | 30.3 |
| 2 | 11.6 | 9.4 | 11.0 | 11.1 |
| 3 | 4.6 | 3.2 | 4.2 | 4.2 |
| 4 + | 3.7 | 2.2 | 3.3 | 2.9 |
| **Nonimputed data** | | | | |
| Smoking in pregnancy, % (n) | 5.8 (1,646/28,252) | 5.2 (1,090/21,149) | 5.2 (698/13,466) | 5.7 (569/10,010) |
| *Missing smoking, n* | *1,152* | *5,397* | *344* | *1,086* |
| Preexisting comorbidities, % (n) | | | | |
| Hypertension | 2.0 (379/19,324) | 1.5 (303/20,162) | 1.3 (119/9,276) | 1.4 (130/9,189) |
| *Missing hypertension, n* | *10,080* | *6,384* | *4,534* | *1,907* |
| Diabetes | 0.9 (162/18,511) | 2.5 (497/20,162) | 1.0 (94/9,153) | 3.4 (299/8,862) |
| *Missing diabetes, n* | *10,893* | *6,384* | *4,657* | *2,234* |
| Systemic Lupus Erythematous | 0.18 (35/19,344) | 0.03 (7/20,154) | 0.17 (16/9,294) | 0.02 (2/8,521) |
| *Missing SLE, n* | *10,060* | *6,392* | *4,516* | *2,575* |
| Antiphospholipid Syndrome | 0.05 (9/19,285) | 0.00 (0/11,629) | 0.05 (5/9,294) | 0.00 (0/4,904) |
| *Missing APS, n* | *10,119* | *14,917* | *4,516* | *6,192* |
| Pregnancy comorbidities, % (n) | | | | |
| Gestational diabetes | 3.5 (833/23,957) | 6.2 (1,242/20,087) | 6.3 (713/11,416) | 8.1 (707/8,699) |
| *Missing GDM, n* | *5,447* | *6,459* | *2,394* | *2,397* |
| Gestational hypertension | 1.7 (308/18,506) | 2.6 (401/15,215) | 1.2 (136/11,418) | 3.4 (219/6,498) |

(*Continued*)

**Table 2.** (Continued)

|  | Prerandomisation period | | Outcome period | |
|---|---|---|---|---|
|  | **Standard Care (n = 29,404)** | **Intervention (GAP) (n = 26,546)** | **Standard Care (n = 13,810)** | **Intervention (GAP) (n = 11,096)** |
| *Missing Gest HT, n* | *10,898* | *11,331* | *2,392* | *4,598* |
| Pre-eclampsia | 0.7 (132/18,504) | 1.8 (368/20,150) | 1.2 (100/8,663) | 2.4 (216/9,185) |
| *Missing Pre-eclampsia, n* | *10,900* | *6,396* | *5,147* | *1,911* |
| Eclampsia | 0.29 (54/18,504) | 0.09 (10/11,372) | 0.30 (26/8,663) | 0.08 (4/4,827) |
| *Missing Eclampsia, n* | *10,900* | *15,174* | *5,147* | *6,269* |
| Infant sex, male, % (n) | 51.3 (15,086/29,397) | 51.3 (13,586/26,494) | 51.1 (7,053/13,798) | 50.7 (5,590/11,023) |
| *Missing Infant sex, n* | *7* | *52* | *12* | *73* |

Data are % (n/N); mean (SD); or median (IQR), unless otherwise specified. Where multiple imputation was used numbers are not provided, only percentages.

APS, Antiphospholipid Syndrome; GAP, Growth Assessment Protocol; GDM, gestational diabetes; Gest HT, gestational hypertension; SLE, Systemic Lupus Erythematous.

description of the full list of ethnicities used for the customised centiles calculator is provided in Tables C and D in S3 Appendix. There were 4 tertiary level clusters in the trial; these were balanced by randomisation of 2 clusters to each of the 2 trial arms.

The proportion of women with an EFW measured by ultrasound after 24 weeks was similar in the intervention and standard care arms during the outcome period (64.0% versus 75.7%; unadjusted difference −11.7%, 95% CI −31.0% to 7.6%; adjusted difference −10.0%, 95% CI −36.2% to 16.1%; adjusted for baseline, age, ethnicity, parity, and stratification factor). In the prerandomisation period, the respective rates were 62.0% versus 43.7% (Table 3). Timing of ultrasound scan after 24 weeks (i.e., utilisation per week of gestation) was broadly similar between trial arms in the outcome period (Fig 2). A common pattern of offering scans at 28, 32, and 36 weeks was observed in both standard care and intervention arms. In the prerandomisation period, a higher proportion of scans at 36 weeks was observed in the intervention arm compared to standard care; no clear difference was observed in other gestations.

The primary outcome of antenatal detection of SGA infants by both customised and population centiles was similar between trial arms (unadjusted difference intervention minus control 1.2%, 95% CI −7.5% to 9.8%; adjusted difference intervention minus control 2.2%, 95% CI −6.4% to 10.7%; adjusted for baseline, age, ethnicity, parity, and stratification factor), as was the test positive rate (unadjusted difference 0.9%, 95% CI −0.6% to 2.5%; adjusted difference 0.8%, 95% CI −0.8% to 2.3%; adjusted for baseline, age, ethnicity, parity, and stratification factor) (Table 4). The association between antenatal detection of SGA at baseline and the comparison period across clusters is displayed in Fig J in S3 Appendix). Measures of diagnostic test performance (antenatal detection, false positive rate, positive predictive value, and negative predictive value) when SGA at birth is defined by customised centiles or by population centiles are provided in Table 4; there were no differences in antenatal detection between trial arms. There were also no differences in the rates of primary and secondary outcomes in most of the prespecified secondary and sensitivity analyses (Tables E, F, and G in S3 Appendix). In the full ITT analysis, the unadjusted difference (intervention minus control) for the primary outcome was −4.0% (95% CI −14.8% to 6.8%), and the adjusted difference was −3.5% (95% CI −14.0% to 7.0%; p = 0.52). There was no difference in the primary outcome in the sensitivity analysis excluding 1 cluster without ultrasound measurement for the prerandomisation period

**Table 3. Utilisation of ultrasound services according to treatment allocation (mITT analysis).**

| | Prerandomisation period | | Outcome period | | Intervention effect size—unadjusted (95% CI) | Intervention effect size—adjusted* (95% CI) | p-value |
|---|---|---|---|---|---|---|---|
| | Standard Care (n = 29,404) | Intervention (GAP) (n = 26,546) | Standard Care (n = 13,810) | Intervention (GAP) (n = 11,096) | | | |
| **Ultrasound** | | | | | | | |
| Proportion of pregnancies with a scan between $18^{+0}$ and $24^{+0}$ weeks, % (n) | 66.2 (19,473/ 29,404) | 82.2 (21,807/ 26,546) | 88.4 (12,212/ 13,810) | 84.2 (9,344/11,096) | −3.7 (−11.6, 4.3) | −10.7 (−36.7, 15.3) | 0.35 |
| Proportion of pregnancies with a scan after $24^{+0}$ weeks, % (n) | 45.1 (13,273/ 29,404) | 60.7 (16,111/ 26,546) | 77.3 (10,677/ 13,810) | 66.1 (7,331/11,096) | −8.4 (−24.9, 8.1) | −12.6 (−32.6, 7.5) | 0.18 |
| Proportion of pregnancies with a scan after $24^{+0}$ weeks with EFW, % (n)† | 43.7 (12,860/ 29,404) | 62.0 (11,629/ 18,751) | 75.7 (10,450/ 13,810) | 64.0 (5,145/8,043) | −11.7 (−31.0, 7.6) | −10.0 (−36.2, 16.1) | 0.35 |
| Number of scans per pregnancy after $24^{+0}$ weeks with EFW, mean (SD) | 0.9 (1.3) | 1.2† (1.3) | 1.5 (1.3) | 1.5† (1.4) | −0.1† (−0.8, 0.6) | −0.2† (−0.6, 0.1) | 0.14 |
| Proportion of pregnancies with no record of ultrasound, % (n) | 27.1 (7,961/ 29,404) | 11.8 (3,121/26,546) | 5.8 (794/13,810) | 9.2 (1,015/11,096) | 2.2 (−5.9, 10.3) | 2.6 (−5.3, 10.6) | 0.45 |

Data are % (n/N) or mean (SD), unless otherwise specified. Effect size provided are differences (intervention minus standard care arm) for the outcome period. 95% CIs and p-values are derived from linear regression where the dependent variable for each outcome was the adjusted cluster summary; p-values are reported only for the adjusted analysis.

CI, confidence interval; EFW, estimated fetal weight using ultrasound; mITT, modified intention to treat.

* Adjusted for baseline, age, ethnicity, parity, and stratification factor.

† Excludes 2 clusters.

(adjusted difference intervention minus control 2.4%, 95% CI −6.1% to 10.8%; p = 0.58); results were in keeping with the main analysis. All minimum requirements for GAP compliance prior to "going live" were met except the e-learning target, which was only met in 1 cluster; therefore, per protocol analysis could not be performed. The intracluster correlation coefficient observed in the outcome period for mITT analysis was 0.008 (95% CI 0.002 to 0.039).

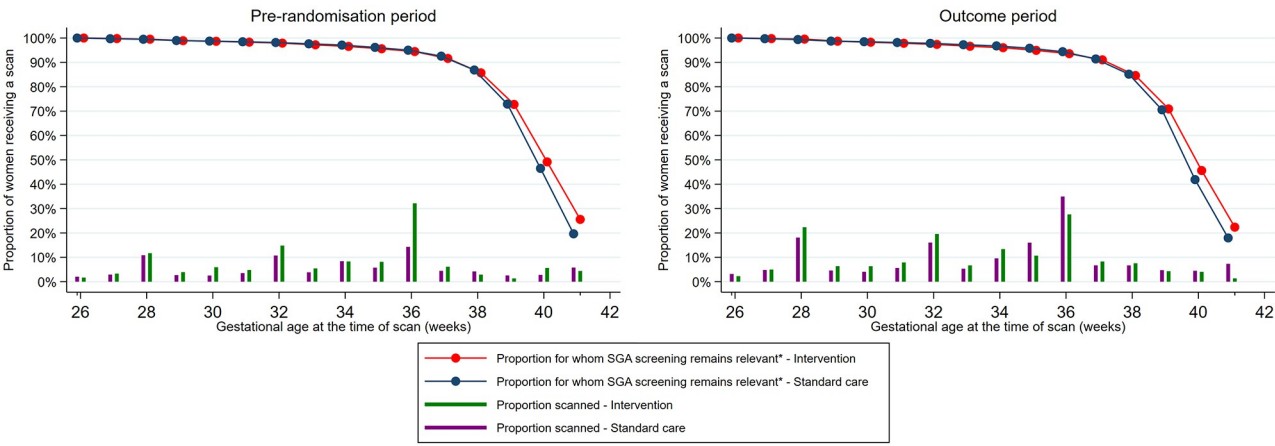

**Fig 2. Ultrasound utilisation per week of gestation in standard care and intervention arms during the prerandomisation and comparison periods.** SGA, small for gestational age. *Pregnancies for which SGA screening remained relevant for each week gestation was defined as ongoing pregnancies (undelivered) that had not been antenatally detected as SGA (growth scans with estimated fetal weight >10th centile or no growth scans) up to that gestational age.

**Table 4. Screening performance according to treatment allocation (mITT analysis).**

| | Prerandomisation | | Outcome period | | Intervention effect size—unadjusted (95% CI) | Intervention effect size—adjusted* (95% CI) | p-value |
|---|---|---|---|---|---|---|---|
| | Standard Care (*n* = 29,404) | Intervention (GAP) (*n* = 26,546) | Standard Care (*n* = 13,810) | Intervention (GAP) (*n* = 11,096) | | | |
| **Primary outcome (SGA by customised and population centiles)** | | | | | | | |
| Proportion of SGA (birthweight), % | 7.2 | 7.6 | 7.2 | 7.6 | - | - | |
| Antenatal detection of SGA, % | 19.1 | 24.4 | 27.7 | 25.9 | 1.2 (−7.5, 9.8) | 2.2 (−6.4, 10.7) | 0.62 |
| Test positive rate, % | 2.4 | 3.2 | 3.4 | 3.7 | 0.9 (−0.6, 2.5)) | 0.8 (−0.8, 2.3) | 0.35 |
| **Secondary outcomes** | | | | | | | |
| **SGA by customised centiles** | | | | | | | |
| Proportion of SGA (birthweight), % | 11.2 | 11.0 | 11.6 | 12.2 | - | - | - |
| Antenatal detection of SGA, % | 14.9 | 19.7 | 21.5 | 22.3 | 2.9 (−3.2, 8.9) | 3.2 (−3.1, 9.4) | 0.32 |
| Specificity†, % | 99.2 | 98.9 | 99.0 | 98.9 | - | - | - |
| Positive predictive value‡, % | 68.9 | 67.1 | 73.3 | 72.9 | - | - | - |
| Negative predictive value‡, % | 90.2 | 90.7 | 90.6 | 89.8 | - | - | - |
| False positive rate†, % | 0.9 | 1.1 | 1.0 | 1.1 | 0.4 −0.4, 1.2) | 0.3 (−0.5, 1.1) | 0.41 |
| False negative rate, % | 85.2 | 80.3 | 78.6 | 77.7 | - | - | - |
| **SGA by population centiles** | | | | | | | |
| Proportion of SGA (birthweight), % | 8.6 | 9.7 | 8.5 | 9.4 | - | - | - |
| Antenatal detection of SGA, % | 17.1 | 21.3 | 25.0 | 21.5 | −0.5 (−9.1, 8.0) | 0.8 (−7.0, 8.7) | 0.83 |
| Specificity†, % | 99.0 | 98.8 | 98.6 | 98.2 | - | - | - |
| Positive predictive value‡, % | 60.9 | 64.2 | 62.5 | 54.8 | - | - | - |
| Negative predictive value‡, % | 92.7 | 91.8 | 93.4 | 92.1 | - | - | - |
| False positive rate†, % | 1.0 | 1.2 | 1.4 | 1.9 | 0.9 (−0.2, 2.1) | 0.8 (−0.3, 1.8) | 0.14 |
| False negative rate, % | 82.9 | 78.7 | 75.0 | 78.5 | - | - | - |

Data are % (n/N), unless otherwise specified. Where multiple imputation was used, numbers are not provided, only percentages. Effect size provided are differences (intervention minus standard care arm) for the outcome period. 95% CIs and *p*-values are derived from linear regression where the dependent variable for each outcome was the adjusted cluster summary; *p*-values are reported only for the adjusted analysis.

CI, confidence interval; GAP, Growth Assessment Protocol; mITT, modified intention to treat; SGA, small for gestational age infant.

* Adjusted for baseline, age, ethnicity, parity, and stratification factor.

† Excludes 1 cluster.

‡ Prerandomisation values exclude 2 clusters, but outcome period excludes only 1 cluster.

There were 2 statistically significant differences among the 26 secondary outcomes explored. When compared to standard care, the intervention was associated with a lower rate of overall stillbirth (unadjusted difference −0.05%, 95% CI −0.21% to 0.11%; adjusted difference −0.07%, 95% CI −0.14% to −0.01%; i.e., 0.7 fewer stillbirths per 1,000 births; adjusted for baseline, age, ethnicity, parity, and stratification factor) and of perinatal mortality (unadjusted difference −0.05%, 95% CI −0.27% to 0.17%; adjusted difference −0.09%, 95% CI −0.17% to −0.004%; i.e., 0.9 fewer perinatal deaths per 1,000 births; adjusted for baseline, age, ethnicity, parity, and stratification factor) (Table 5). The post hoc sensitivity analysis of stillbirth led to an unadjusted odds ratio (95% CI) for the intervention effect of 1.30 (95% CI 0.68 to 2.47), and adjusted odds ratio of 0.77 (95% CI 0.30 to 1.99); we do not attempt to reexpress this effect as a difference between arms as the methodology to do so with imputed data is not yet established.

In the subgroup analysis of outcomes for SGA infants (defined by both population and customised centiles; *n* = 1,802 pregnancies of which 31 were stillborn), SGA infants in the intervention arm were born 2 days earlier, had a lower mean birthweight, and lower rates of

**Table 5. Secondary clinical outcomes according to treatment allocation (mITT analysis).**

| | Prerandomisation period | | Outcome period | | Intervention effect size —unadjusted (95% CI) | Intervention effect size —adjusted* (95% CI) | p-value |
|---|---|---|---|---|---|---|---|
| | Standard Care (*n* = 29,404) | Intervention (GAP) (*n* = 26,546) | Standard Care (*n* = 13,810) | Intervention (GAP) (*n* = 11,096) | | | |
| **Maternal outcomes** | | | | | | | |
| Induction of labour, % | 25.1 | 26.3 | 26.9 | 29.5 | 2.8 (−4.2, 9.8) | 1.7 (−0.4, 3.8) | 0.11 |
| Mode of birth, % | | | | | | | |
| *Spontaneous vaginal delivery* | 58.1 | 58.7 | 54.5 | 54.0 | 1.5 (−4.5, 7.5) | -0.1 (−2.6, 2.4) | 0.94 |
| *Operative vaginal delivery* | 13.7 | 15.3 | 14.1 | 14.4 | 0.3 (−3.1, 3.6) | -0.1 (−1.6, 1.4) | 0.87 |
| *Elective cesarean section* | 12.3 | 12.2 | 13.9 | 14.6 | −0.9 (−5.7, 3.8) | −0.6 (−1.5, 0.4) | 0.24 |
| *Emergency cesarean section* | 15.6 | 13.6 | 17.2 | 16.7 | −0.8 (−4.4, 2.8) | 0.6 (−1.6, 2.8) | 0.59 |
| Postpartum haemorrhage (>1,500 mls), % | 2.7 | 2.3 | 2.7 | 2.5 | −0.4 (−1.2, 0.3) | −0.1 (−0.5, 0.3) | 0.66 |
| Third/fourth degree tears, % | 2.2 | 2.4 | 1.9 | 1.8 | 0.0 (−0.8, 0.7) | −0.1 (−0.6, 0.4) | 0.78 |
| Epidural, % | 36.5 | 27.9 | 36.4 | 28.2 | −13.0 (−33.7, 7.7) | 5.6 (−1.4, 12.7) | 0.12 |
| Episiotomy, % | 17.7 | 23.1 | 17.6 | 21.8 | 16.4 (−10.1, 43.0) | −2.3 (−6.4, 1.9) | 0.28 |
| **Neonatal outcomes** | | | | | | | |
| Gestational age at birth, weeks mean (SD) | 39.5 (2.0) | 39.5 (2.0) | 39.4 (1.9) | 39.4 (2.0) | −0.1 (−0.2, 0.1) | 0.0 (−0.1, 0.1) | 0.80 |
| *Preterm birth (<37 weeks), %* | 5.6 | 6.0 | 6.1 | 6.4 | 0.3 (−1.3, 1.8) | 0.0 (−0.8, 0.9) | 0.94 |
| Birthweight (g), mean (SD) | 3,348 (559) | 3,325 (558) | 3,326 (552) | 3,297 (567) | −24.1 (−87.2, 39.0) | −7.7 (−21.9, 6.4) | 0.28 |
| *Condition at birth* | | | | | | | |
| Apgar score <7 at 5 minutes, % | 2.0 | 1.9 | 2.2 | 1.7 | −0.5 (−1.1, 0.1) | −0.2 (−0.4, 0.1) | 0.29 |
| Arterial cord pH <7.1, % | 2.3 | 2.8 | 2.0 | 2.9 | 0.7 (−1.0, 2.4) | 0.3 (−0.4, 1.0) | 0.44 |
| Respiratory support at birth, % | 4.4 | 6.3 | 4.1 | 4.8 | 1.2 (−3.5, 5.8) | −1.0 (−2.7, 0.7) | 0.26 |
| *Neonatal admissions* | | | | | | | |
| Neonatal unit admission (inc HDU and SCBU), % | 14.9 | 8.1 | 16.2 | 7.4 | −8.3 (−27.5, 10.8) | 0.4 (−0.8, 1.7) | 0.48 |
| *Major neonatal morbidity* | | | | | | | |
| Any major neonatal morbidity, % | 4.5 | 6.2 | 5.5 | 4.7 | −1.5 (−4.9, 1.8) | −1.2 (−3.4, 1.0) | 0.28 |
| *Any neonatal brain injury (HIE + IVH), %* | 0.44 | 0.44 | 0.41 | 0.34 | | | |
| *Supplementary O$_2$ >28 days, %* | 0.16 | 0.16 | 0.09 | 0.15 | | | |
| *Necrotising enterocolitis, %* | 0.18 | 0.15 | 0.12 | 0.08 | | | |
| *Sepsis, %* | 4.50 | 6.13 | 5.37 | 4.60 | | | |
| *Retinopathy of prematurity, %* | 0.11 | 0.12 | 0.17 | 0.06 | | | |
| *Minor Neonatal morbidity* | | | | | | | |
| Any minor neonatal morbidity, % | 2.8 | 4.5 | 2.6 | 3.0 | 0.5 (−1.4, 2.4) | −0.0 (−1.6, 1.5) | 0.96 |
| *Hypothermia, %* | 0.14 | 0.41 | 0.17 | 0.14 | | | |
| *Hypoglycaemia, %* | 1.43 | 1.72 | 1.19 | 0.86 | | | |

*(Continued)*

**Table 5.** (Continued)

| | Prerandomisation period | | Outcome period | | Intervention effect size—unadjusted (95% CI) | Intervention effect size—adjusted* (95% CI) | p-value |
|---|---|---|---|---|---|---|---|
| | Standard Care (n = 29,404) | Intervention (GAP) (n = 26,546) | Standard Care (n = 13,810) | Intervention (GAP) (n = 11,096) | | | |
| *Nasogastric feeding, %* | *2.37* | *3.62* | *1.98* | *2.62* | | | |
| **Perinatal loss** | | | | | | | |
| Stillbirth, % | 0.30 | 0.40 | 0.36 | 0.31 | −0.05 (−0.21, 0.11) | −0.07 (−0.14, −0.01) | 0.03 |
| Neonatal death, % | 0.07 | 0.13 | 0.04 | 0.07 | 0.01 (−0.08, 0.10) | −0.02 (−0.08, 0.04) | 0.56 |
| Perinatal mortality, % | 0.37 | 0.49 | 0.41 | 0.37 | −0.05 (−0.27, 0.17) | −0.09 (−0.17, −0.004) | 0.04 |

Data are % (n/N) or mean (SD), unless otherwise specified. Where multiple imputation was used, numbers are not provided, only percentages Effect size provided are differences (intervention minus standard care arm) for the outcome period. 95% CIs and *p*-values are derived from linear regression where the dependent variable for each outcome was the adjusted cluster summary; *p*-values are reported only for the adjusted analysis.

CI, confidence interval; GAP, Growth Assessment Protocol, HDU, high dependence unit; HIE, hypoxic ischemic injury; IVH, intraventricular haemorrhage; mITT, modified intention to treat; SCBU, special care baby unit; $O_2$, oxygen.

* Adjusted for baseline, age, ethnicity, parity, and stratification factor.

stillbirth compared to SGA infants from standard care (Table 6). There were no differences in other neonatal or maternal outcomes in the subgroup analysis, including preterm birth (<37 weeks; Table 6) and late preterm birth ($34^{+0}$ to $36^{+6}$ weeks; post hoc analysis, 9.1% versus 8.4% for intervention and standard care arms, respectively; adjusted difference 0.3%, 95% CI −1.9% to 2.6%). The change in mean gestational age at birth reflects fewer SGA babies born at or after 39 weeks in the intervention arm compared to standard care arm (post hoc analysis, 56.3% versus 61.2%; adjusted difference −8.3%, 95% CI −14.9% to −1.7%). Clinical outcomes using available case data and for women with a scan recorded in the cluster between $18^{+0}$ and $24^{+0}$ weeks are reported in Tables H and I in S3 Appendix, respectively).

Assessment of implementation (fidelity, dose, and reach) of GAP was performed at all implementing clusters. Implementing sites had guidelines in which concordance to the Perinatal Institute guidance ranged from high to low. All clusters achieved the face-to-face training target, but only 1 cluster achieved the e-learning target. Of the 595 women whose maternity records were reviewed, 84.9% were correctly risk stratified according to GAP guidelines (range between clusters 78.6% to 87.5%) and 88.7% had a GROW chart in their notes (range between clusters 62.2% to 98.3%). Intervention dosage varied; 30.7% (range between clusters 8.2% to 53.2%) of low-risk women had at least the minimum recommended number of fundal height measurements plotted on their GROW chart and 8.5% (range between clusters 0.0% to 16.7%) of women with risk factors for SGA had at least the minimum number of growth scans as recommended by GAP (Table 7). Detailed qualitative data with clinicians and other staff exploring implementation will be reported separately. In the standard care arm, there was wide variation in term of guidance for screening for SGA including variation in timing and interpretation of fundal height measurement, factors indicating high-risk status and number and frequency of ultrasound for high-risk women.

## Discussion

The DESiGN trial has found that GAP was not superior to standard care for the antenatal detection of SGA, confirmed at birth by both population and customised centiles. All intervention clusters achieved the preimplementation requirements for access to GROW software, except for the e-learning target. In intervention clusters, GAP was implemented with varied

**Table 6. Subgroup analysis: Clinical outcomes among SGA babies by population and customised according to treatment allocation (mITT analysis).**

| | Prerandomisation period | | Outcome period | | Intervention effect size —unadjusted (95% CI) | Intervention effect size —adjusted* (95% CI) | p-value |
|---|---|---|---|---|---|---|---|
| | Standard Care (n = 2,134) | Intervention (GAP) (n = 1,932) | Standard Care (n = 995) | Intervention (GAP) (n = 807) | | | |
| **Maternal outcomes** | | | | | | | |
| Induction of labour, % | 33.0 | 32.2 | 35.1 | 36.8 | 2.1 (−9.2, 13.3) | 3.6 (−2.6, 9.8) | 0.25 |
| Mode of birth, % | | | | | | | |
| *Spontaneous vaginal delivery* | *54.2* | *52.2* | *53.6* | *48.7* | *−3.5 (−12.4, 5.3)* | *−3.3 (−7.4, 0.7)* | *0.11* |
| *Operative vaginal delivery* | *13.2* | *15.8* | *13.2* | *16.1* | *3.7 (−1.3, 8.7)* | *0.6 (−3.0, 4.2)* | *0.75* |
| *Elective cesarean section* | *9.2* | *10.3* | *10.6* | *9.4* | *−1.6 (−6.3, 3.1)* | *−1.7 (−5.1, 1.6)* | *0.32* |
| *Emergency cesarean section* | *22.6* | *21.0* | *21.4* | *25.3* | *2.3 (−4.6, 9.2)* | *2.4 (−0.9, 5.8)* | *0.16* |
| Postpartum haemorrhage (>1,500 mls), % | 1.3 | 1.4 | 1.3 | 1.7 | 0.4 (−0.6, 1.4) | 0.4 (−0.5, 1.3) | 0.39 |
| Third/fourth degree tears, % | 0.9 | 1.1 | 0.8 | 1.4 | 1.1 (0.0, 2.3) | 1.1 (−0.1, 2.2) | 0.08 |
| Epidural, % | 36.9 | 30.6 | 36.5 | 29.4 | −12.9 (−31.3, 5.5) | −0.8 (−6.7, 5.0) | 0.78 |
| Episiotomy, % | 17.0 | 25.3 | 18.2 | 22.9 | 17.6 (−6.0, 41.1) | −4.0 (−8.9, 0.9) | 0.11 |
| **Neonatal outcomes** | | | | | | | |
| Gestational age at birth, weeks mean (SD) | 38.9 (3.0) | 39.0 (2.9) | 38.8 (3.0) | 38.6 (3.1) | −0.3 (−0.6, 0.1) | −0.3 (−0.5, −0.1) | 0.002 |
| *Preterm birth (<37 weeks), %* | *13.8* | *12.8* | *13.7* | *16.5* | *3.0 (−1.92, 7.9)* | *2.3 (−1.5, 6.2)* | *0.23* |
| Birthweight (g), mean (SD) | 2,492 (534) | 2,496 (532) | 2,482 (550) | 2,436 (550) | −55 (−131, 21) | −58 (−99, −18) | 0.005 |
| ***Condition at birth*** | | | | | | | |
| Apgar score <7 at 5 minutes, % | 4.4 | 5.0 | 5.2 | 4.1 | −0.9 (−2.53, 0.7) | −0.5 (−1.7, 0.8) | 0.45 |
| Arterial cord pH <7.1, % | 3.1 | 3.8 | 2.8 | 3.7 | 0.6 (−1.9, 3.0) | −0.3 (−1.4, 0.8) | 0.58 |
| Respiratory support at birth, % | 8.6 | 10.6 | 7.1 | 8.0 | 1.6 (−4.3, 7.5) | −1.3 (−4.4, 1.8) | 0.40 |
| ***Neonatal admissions*** | | | | | | | |
| Neonatal unit admission (inc HDU and SCBU), % | 25.2 | 17.0 | 22.9 | 15.0 | −5.3 (−23.4, 12.8) | 1.5 (−2.4, 5.4) | 0.46 |
| ***Major neonatal morbidity*** | | | | | | | |
| Any major neonatal morbidity, % | 8.4 | 11.5 | 8.4 | 9.2 | 0.0 (−4.3, 4.3) | 0.5 (−3.2, 4.2) | 0.80 |
| *Any neonatal brain injury (HIE + IVH), %* | *1.31* | *1.03* | *0.89* | *1.15* | | | |
| *Supplementary $O_2$ >28 days, %* | *0.66* | *0.63* | *0.39* | *0.46* | | | |
| *Necrotising enterocolitis, %* | *0.94* | *0.99* | *0.19* | *0.35* | | | |
| *Sepsis, %* | *8.27* | *11.33* | *8.21* | *8.95* | | | |
| *Retinopathy of prematurity, %* | *0.39* | *0.33* | *0.21* | *0.10* | | | |
| ***Minor neonatal morbidity*** | | | | | | | |
| Any minor neonatal morbidity, % | 8.9 | 12.6 | 6.3 | 8.8 | 3.2 (−1.9, 8.3) | 1.9 (−3.1, 6.9) | 0.46 |
| *Hypothermia, %* | *0.71* | *2.16* | *0.90* | *0.98* | | | |
| *Hypoglycaemia, %* | *5.62* | *5.89* | *3.38* | *3.18* | | | |
| *Nasogastric feeding, %* | *7.76* | *10.77* | *5.21* | *7.39* | | | |
| ***Perinatal loss*** | | | | | | | |
| Stillbirth, % | 1.67 | 1.86 | 2.19 | 1.39 | −1.03 (−1.88, −0.18) | −0.76 (−1.50, −0.03) | 0.04 |

*(Continued)*

**Table 6.** (*Continued*)

| | Prerandomisation period | | Outcome period | | Intervention effect size —unadjusted (95% CI) | Intervention effect size —adjusted* (95% CI) | *p*-value |
|---|---|---|---|---|---|---|---|
| | Standard Care (*n* = 2,134) | Intervention (GAP) (*n* = 1,932) | Standard Care (*n* = 995) | Intervention (GAP) (*n* = 807) | | | |
| Neonatal death, % | 0.36 | 0.68 | 0.18 | 0.38 | 0.04 (−0.44, 0.52) | −0.11 (−0.60, 0.38) | 0.67 |
| Perinatal mortality, % | 2.04 | 2.24 | 2.37 | 1.77 | −0.99 (−2.06, 0.08) | −0.69 (−1.47, 0.09) | 0.08 |

Data are % (n/N) or mean (SD), unless otherwise specified. Where multiple imputation was used, numbers are not provided, only percentages. Effect size provided are differences (intervention minus standard care arm) for the outcome period. 95% CIs and *p*-values are derived from linear regression where the dependent variable for each outcome was the adjusted cluster summary; *p*-values are reported only for the adjusted analysis.

CI, confidence interval; GAP, Growth Assessment Protocol; HDU, high dependence unit; HIE, hypoxic ischemic injury; IVH, intraventricular haemorrhage; mITT, modified intention to treat; SCBU, special care baby unit; $O_2$, oxygen.

* Adjusted for baseline, age, ethnicity, parity, and stratification factor.

levels of fidelity (high rates of face-to-face training, varied concordance of cluster site guidelines with GAP, high concordance with GAP risk stratification protocols), high levels of reach (majority of women had a GROW chart), but variable dose (low number of fundal height measurements plotted, number of growth scans below that which is recommended by GAP, high rates of referral for suspected SGA).

**Table 7. Assessment of implementation strength: reach, dose, and fidelity.**

| Implementation Outcome | Measure of outcome | Overall results | Median cluster score (range) |
|---|---|---|---|
| **Fidelity** (*the extent to which core components were consistently implemented*) | Concordance of cluster guidelines for SGA detection to those recommended in GAP | High fidelity (2 clusters), medium fidelity (2 clusters), low fidelity (1 cluster)* | |
| | >75% of staff members from each professional group (midwives, sonographers, obstetricians) trained in face-to-face methods | All 5 clusters compliant | |
| | >75% of staff members from each professional group (midwives, sonographers, obstetricians) trained using e-learning methods | One cluster met training target (4 clusters did not). | |
| | Proportion of women risk stratified according to GAP guidelines | 84.9%† (505/595) | 84.2% (78.6%–87.5%) |
| **Reach** (*participation in the intervention by clinicians*) | Proportion of women with a GAP-GROW chart in the notes | 88.7% (528/595) | 94.2% (62.2%–98.3%) |
| **Dose** (*proportion of each component delivered*) | Number of fundal heights plotted for low risk women, median (IQR)‡ | 3 (2–4) | 3 (1–4) |
| | Proportion of low-risk women with a GROW chart who had at least the minimum expected number of fundal height measurements performed and plotted on GROW‡ | 30.7% (114/371) | 31.4% (8.2%–53.2%) |
| | Proportion of low-risk women referred for growth scan when definite plot deviation‡ | 74.2%§ (69/102) | 66.7% (40.0%–80.9%) |
| | Number of fetal growth ultrasound scans completed for high-risk women, median (IQR)‡ | 3 (2–4) | 3 (2–4) |
| | Proportion of high-risk women with a GROW chart who had at least the minimum expected number of growth scans performed and plotted on GROW‡ | 8.5% (17/201) | 5.3% 0(0.0%–16.7%) |

Data are % (n/N) or median (IQR).

GAP, Growth Assessment Protocol; GROW, Gestation-Related Optimal Weight chart; SGA, small for gestational age.

* High fidelity (only occasional differences where GAP recommendations were partially included); medium fidelity (with partial or no inclusion of GAP recommendations in less than half of the recommendations); low fidelity (with partial or no inclusion of GAP recommendations throughout the guidelines, affecting over half of the recommendations).

† Around 18/90 women who were not correctly risk stratified by GAP guidelines were correctly risk stratified according to local policy.

‡ Risk status is as classified by clinician at booking.

§ Approximately 11.2% (16/102) additional women did have a growth scan, but documented as another indication, e.g., reduced fetal movements.

To the best of our knowledge, the DESiGN trial is the first randomised control trial that compared the effect of GAP and standard care on the ultrasound-detection of SGA. The intervention was not superior to standard care when implemented in this study setting. It is important to note that at the time of the DESiGN trial, there was concurrent national implementation of the "Saving Babies' Lives' care bundle, which aimed to reduce rates of stillbirth through 4 components (smoking cessation, risk assessment for and surveillance of fetal growth restriction, raising awareness of reduced fetal movements, and effective fetal monitoring during labour) [18]; this has been shown to increase use of ultrasound and improve the detection of SGA [25]. The outcome period of this trial was in 2018/2019, at least 2 years after the implementation of the care bundle. While the NHS England and NHS Improvement (London) Clinical Leadership Group exempted the 5 London-based clusters in the standard care arm of this study from implementing the care bundle component related to fetal growth restriction during the study period, most units chose to implement at least some of the care bundle strategies. In previous observational studies reporting increased antenatal detection of SGA or reduced stillbirth following GAP implementation, preimplementation groups were not affected by this care bundle. This may explain some of the differences observed in antenatal detection of SGA between this and previous studies; the different study design between this randomised control trial and previous studies, which were all observational, may also explain the different results observed.

Our process evaluation highlights variation in implementation of GAP, which was also reported in the SPiRE Study [25], where 15 of 19 included maternity units had implemented GAP. The SPiRE study group found that most of the 15 local guidelines collected from GAP-implementing sites were only partially compliant with 4 out of 5 components that feature both in the fetal growth restriction element of the Saving Babies' Lives care bundle and in GAP guidelines [26]. We also observed partial concordance with GAP guidelines in this trial, demonstrated through variable implementation fidelity.

In England, multiparous women are routinely offered fewer antenatal appointments than required for compliance with GAP fundal height measurement frequency, this may partly explain why the number of fundal heights plotted is lower than that recommended by GAP (every 3 weeks). Implementation dose in terms of number of scans conducted for each woman at high risk of SGA was lower than that which is recommended by GAP (3 versus 4 scans for women with term birth). This may be explained by common practice in England whereby serial growth scans are offered at 28, 32, and 36 weeks, rather than 3-weekly. Indeed, post hoc exploration of implementation dose data has shown that 74% of high-risk women in the intervention arm of this study had 2 or more growth scans after 24 weeks, suggesting a less frequent surveillance programme than recommended by GAP. The exploratory analysis of timing of ultrasound utilisation requested by the reviewers/academic editor also supports this hypothesis and describe a similar surveillance pattern in the standard care arm. The costs related to GAP include both the annual charge from the Perinatal Institute to access the programme, training costs, and any potential increase in use of clinical resources; these need to be considered when evaluating utility of GAP. A detailed economic analysis will be reported separately.

We observed a lower rate of overall stillbirth and perinatal mortality, as well as SGA stillbirth in the intervention arm compared to standard care arm during the outcome period. The fact that this was not achieved though the expected pathway of improving detection of SGA at birth, our primary outcome, does raise the possibility of a chance finding, and the finding was not confirmed in the (albeit post hoc) sensitivity analysis. Although we are limited in our ability to ascertain the drivers of this potential effect, it is plausible that the lower proportion of births at or after 39 weeks observed among SGA babies in the intervention arm may have mediated this effect. There is conflicting evidence regarding the benefit of offering earlier

iatrogenic birth to women with SGA fetuses as while it may prevent stillbirth/perinatal mortality [27], adversely, it may increase rates of short-term neonatal morbidity and poorer developmental outcomes in childhood [28,29]. Complex interventions such as GAP may have effects that do not necessarily lie on the expected pathway; however, we note the need to replicate these findings before they can be considered robust given the number of secondary outcomes in this study.

We have not performed statistical testing to assess for changes between prerandomisation and outcome period as per prespecified analysis plan; however, we did observe some differences. In particular, the use of ultrasound seems to have markedly increased during the study in standard care clusters, which likely relates to the rollout of the Saving Babies' Lives care bundle, at least in part. The SPiRe Study reported increased utilisation of ultrasound with implementation of the care bundle; the association was related to the overall care bundle and not to any specific component. Despite exempt from the fetal growth restriction component of the care bundle, clusters in this trial may have increased the utilisation of ultrasound by other related strategies such as the reduced fetal movements component.

The antenatal detection of neonates confirmed to be SGA at birth by customised centiles (secondary outcome) in this study was not higher in the intervention arm, which suggests the choice of growth chart may have limited influence in detection of SGA. Previous observational studies explored the value of customised centiles alone (not as part of GAP). We recognise that these studies have reported that population and customised charts have similar performance in detecting adverse perinatal outcomes after accounting for false positive rates for term births [30] and that the stronger associations between customised centiles and adverse perinatal outcomes (when compared to population centiles) were explained by confounding with preterm birth and maternal obesity [31], even though this is challenged by other authors.

The strength of this study is that, to the best of our knowledge, it is the first randomised trial assessing the effect of the GAP. DESiGN was a pragmatic trial capturing the real-life challenges of implementing complex interventions into clinical care and included a robust process evaluation and examination of implementation strength and variability. The trial has primarily used data from routinely collected electronic patient records, which has allowed cost-efficient inclusion of data from a large number of pregnancies. The primary outcome was antenatal ultrasound detection of SGA (after 24 completed weeks). We defined this as infants who are SGA (i.e., birthweight less than 10th centile) according to (i) population (UK1990 birthweight centiles) and (ii) customised (GROW) charts; this is considered to identify those at highest risk of adverse perinatal outcomes [32]. This is an important strength as both GAP and standard care target the detection of these infants.

We were unable to assess the impact of complete attainment of the GAP preimplementation requirements because only 1 implementing cluster achieved the training target for e-learning. The optimal interval between commencing GAP use and assessment of its effect is unknown. This study had a median interval of 9 months (range 6 to 12) from antenatal booking of women with the opportunity of exposure to GAP until commencement of outcome data collection. While the learning process of care providers may delay full programme effectiveness, an alternative "pioneering effect" may be working in the opposite direction [33]. Other limitations include issues related to the availability, or format, of data that are inherent in the use of routinely collected data, though we followed clear protocols in harmonisation and linkage of data from multiple electronic systems to minimise any variations in data quality between the randomised arms [15]. Missingness for characteristics (including customisation factors) was dealt with by multiple imputation, which is dependent on the assumption that results after inclusion of variables in the imputation model will be consistent between those with and without missing data. It is unlikely that randomisation to GAP or standard care would alter

completeness of routine data collection in any cluster; therefore, this assumption is likely to be met. Ethnicity documented in hospital systems was often not as granular as that required by the customised calculator. One prespecified subgroup analysis exploring the effect of intervention in women stratified as high risk and low risk separately was not possible given lack of detailed data on some risk factors used to stratify women. The number of units randomised was modest and power was somewhat reduced by the failure of 2 units to contact the provider of GAP leading to their exclusion from our main analyses; however, the observed intracluster correlation coefficient was lower than that assumed for the power calculation; this would have preserved power to some extent.

We are not aware of other studies of GAP implementation that report as detailed assessment of the standardised implementation outcomes (fidelity, reach, and dose) as that performed in this trial [19], and by which we can benchmark these findings. While it is possible that the variable dose of implementation may explain the results of this trial, DESiGN was a pragmatic trial intended to reflect implementation in the real world. It is therefore possible that the implementation variability seen in this trial reflects the reality of implementing a complex intervention in a health service with competing needs on resources. A recent observational study of GAP implementation across the UK also described variation in implementation using nonstandardised outcomes. Their analysis demonstrated a greater reduction of stillbirth rates in maternity units that had completely implemented GAP (defined by reporting the birthweight and outcomes of more than 75% of births via the GAP online tool) compared with those that did not implement GAP [34]. A third of maternity units (31%; $n = 29/94$) implementing GAP achieved only partial implementation. The rate of stillbirth was no different between maternity units with partial or no implementation of GAP. The collective evidence from these studies highlights the challenges and variation in implementation of GAP.

This pragmatic study provides the only evidence from a randomised control trial regarding the effect of GAP, to the best of our knowledge. The GAP programme was not superior to standard care in the detection of SGA at birth by both population and customised centiles in this setting. Given the variable implementation observed, it is imperative that future studies assessing implementation of GAP or other interventions to improve perinatal outcomes, use standardised implementation outcomes (fidelity, reach, and dose) in order to determine the generalisability of our findings, identify barriers to implementation, and hence better inform policy for improving perinatal outcomes.

### Dissemination to participants and related patient and public communities

Participating institutions and maternity units will be informed of the results soon after acceptance and any embargo period. We expect participating maternity units to share results locally in their communities aiming to also reach women that were pregnant during the study period. We will communicate with relevant stakeholders including SANDS and Tommy's Charities. The main results of the current research will also be disseminated to related patients and the public through blogs, press releases, newspapers, and conferences.

### Supporting information

**S1 Protocol. The DESiGN trial Study Protocol.**
(DOCX)

**S1 Appendix. Statistical analysis plan.**
(DOCX)

**S2 Appendix. Additional methodology.**
(DOCX)

**S3 Appendix. Supplementary tables and figures.**
(DOCX)

**S1 CONSORT Checklist. CONSORT Statement Checklist.**
(DOCX)

## Acknowledgments

We would like to thank the members of the DESiGN Collaborative Group for their contribution to this study: Spyros Bakalis, Claire Rozette and Marcelo Canda (from Guy's and St Thomas' Hospital NHS Foundation Trust), Simona Cicero, Olayinka Akinfenwa, Philippa Cox and Lisa Giacometti (from Homerton University Hospital NHS Foundation Trust), Elisabeth Peregrine, Lyndsey Smith and Sam Page (from Kingston Hospital NHS Foundation Trust), Deepa Janga and Sandra Essien (from North Middlesex University Hospital NHS Trust), Renata Hutt (from Royal Surrey County Hospital NHS Foundation Trust), Yaa Acheampong, Bonnie Trinder and Louise Rimell (from St George's University Hospitals NHS Foundation Trust), Janet Cresswell and Sarah Petty (from Chesterfield Royal Hospital NHS Foundation Trust), Bini Ajay, Hannah O'Donnell and Emma Wayman (from Croydon Health Services NHS Trust), Mandish Dhanjal, Muna Noori, and Elisa Iaschi (from Imperial College Healthcare NHS Trust), Raffaele Napolitano, Iris Tsikimi and Rachel Das (from University College London Hospitals NHS Foundation Trust), Fiona Ghalustians and Francesca Hanks (from Chelsea and Westminster Hospital NHS Foundation Trust), Laura Camarasa (from Hillingdon Hospitals NHS Foundation Trust), Hiran Samarage and Stephen Hiles (from London North West Healthcare NHS Trust). We would also like to thank the DESiGN Trial Steering Commetee/Data Monitoring Committee members: Anna David (from University College London), David Howe (from University Hospital Southampton), Nadine Seward (from King's College London), Elizabeth Allen (from the London School of Hygiene and Tropical Medicine), and Jillian Francis (from The University of Melbourne). At last, we wish to thank the Stillbirth Clinical Study Group and the Royal College of Obstetricians and Gynaecologists for reviewing the study protocol during development of the study.

The views expressed are those of the author[s] and not necessarily those of the NIHR, the Department of Health and Social Care, or any of the other listed funders.

## Author Contributions

**Conceptualization:** Matias C. Vieira, Andrew Healey, Kirstie Coxon, Alessandro Alagna, Donald Peebles, Neil Marlow, Lesley McCowan, Christoph Lees, Deborah A. Lawlor, Asma Khalil, Jane Sandall, Andrew Copas, Dharmintra Pasupathy.

**Data curation:** Matias C. Vieira, Sophie Relph, Maria Elstad, Bolaji Coker, Natalie Moitt, Andrew Healey, Kirstie Coxon, Jane Sandall, Andrew Copas, Dharmintra Pasupathy.

**Formal analysis:** Matias C. Vieira, Sophie Relph, Walter Muruet-Gutierrez, Natalie Moitt, Andrew Healey, Kirstie Coxon, Jane Sandall, Andrew Copas, Dharmintra Pasupathy.

**Funding acquisition:** Asma Khalil, Jane Sandall, Dharmintra Pasupathy.

**Investigation:** Matias C. Vieira, Sophie Relph, Walter Muruet-Gutierrez, Maria Elstad, Bolaji Coker, Natalie Moitt, Louisa Delaney, Chivon Winsloe, Andrew Healey, Kirstie Coxon, Alessandro Alagna, Annette Briley, Mark Johnson, Louise M. Page, Donald Peebles,

Andrew Shennan, Baskaran Thilaganathan, Neil Marlow, Lesley McCowan, Christoph Lees, Deborah A. Lawlor, Asma Khalil, Jane Sandall, Andrew Copas, Dharmintra Pasupathy.

**Methodology:** Matias C. Vieira, Sophie Relph, Kirstie Coxon, Christoph Lees, Deborah A. Lawlor, Asma Khalil, Jane Sandall, Andrew Copas, Dharmintra Pasupathy.

**Project administration:** Matias C. Vieira, Sophie Relph, Louisa Delaney, Chivon Winsloe, Andrew Healey, Kirstie Coxon, Asma Khalil, Jane Sandall, Dharmintra Pasupathy.

**Supervision:** Dharmintra Pasupathy.

**Writing – original draft:** Matias C. Vieira, Sophie Relph, Andrew Copas, Dharmintra Pasupathy.

**Writing – review & editing:** Matias C. Vieira, Sophie Relph, Walter Muruet-Gutierrez, Maria Elstad, Bolaji Coker, Natalie Moitt, Louisa Delaney, Chivon Winsloe, Andrew Healey, Kirstie Coxon, Alessandro Alagna, Annette Briley, Mark Johnson, Louise M. Page, Donald Peebles, Andrew Shennan, Baskaran Thilaganathan, Neil Marlow, Lesley McCowan, Christoph Lees, Deborah A. Lawlor, Asma Khalil, Jane Sandall, Andrew Copas, Dharmintra Pasupathy.

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
