## [Editor Report · Decision Letter 0]

15 Sep 2021

Dear Dr Vieira, 

Thank you for submitting your manuscript entitled "Effect of the Growth Assessment Protocol (GAP) on the detection of small for gestational age: the DESiGN cluster randomised trial." for consideration by PLOS Medicine.

Your manuscript has now been evaluated by the PLOS Medicine editorial staff as well and I am writing to let you know that we would like to send your submission out for external peer review. Before doing so, we will require clarification regarding the start date of the trial (specified as November 2015 in the Abstract, and November 2016 in the Methods) since PLOS Medicine requires that all trials are prospectively registered. It would be appreciated if you could contact me directly by email lgaynor@plos.org with this information as soon as possible. 

Before we can send your manuscript to reviewers, we also need you to complete your submission by providing the metadata that is required for full assessment. To this end, please login to Editorial Manager where you will find the paper in the 'Submissions Needing Revisions' folder on your homepage. Please click 'Revise Submission' from the Action Links and complete all additional questions in the submission questionnaire.

Please re-submit your manuscript within two working days, i.e. by Sep 17 2021 11:59PM.

Kind regards,

Louise Gaynor-Brook, MBBS PhD

Associate Editor, PLOS Medicine

---

## [Decision Letter · Decision Letter 1]

27 Oct 2021

Dear Dr. Vieira,

Thank you very much for submitting your manuscript "Effect of the Growth Assessment Protocol (GAP) on the detection of small for gestational age: the DESiGN cluster randomised trial." (PMEDICINE-D-21-03941R1) for consideration at PLOS Medicine. 

Your paper was evaluated by four independent reviewers, including a statistical reviewer, and was discussed among all the editors here and with an academic editor with relevant expertise. The reviews are appended at the bottom of this email and any accompanying reviewer attachments can be seen via the link below:

[LINK]

In light of these reviews, I am afraid that we will not be able to accept the manuscript for publication in the journal in its current form, but we would like to consider a revised version that addresses the reviewers' and editors' comments. Obviously we cannot make any decision about publication until we have seen the revised manuscript and your response, and we plan to seek re-review by one or more of the reviewers. 

We expect to receive your revised manuscript by Nov 17 2021 11:59PM. Please email us (plosmedicine@plos.org) if you have any questions or concerns.

We look forward to receiving your revised manuscript. 

Sincerely,

Louise Gaynor-Brook, MBBS PhD

PLOS Medicine

plosmedicine.org

Comments from the Academic Editor:

This was a very challenging study and unfortunately has limited implications for practice due to the significant logistical challenges. That said it includes some very important messages and articulates many of the challenges associated with complex intervention implementation, and there are important lessons for future studies in this area. There are several issues highlighted by the reviewers which need additional attention - particularly the issue around standard care and the fact that ultrasounds use went up only in the standard care arm. Further detail on the timing of USS scan and detail on acquisition and more importantly validation of the primary outcome is also required.

General comments:

Throughout the paper, please adapt reference call-outs to the following style: "... for high income countries [2,3]." (noting the absence of spaces within the square brackets).

Data availability:

PLOS Medicine requires that the de-identified data underlying the specific results in a published article be made available, without restrictions on access, in a public repository or as Supporting Information at the time of article publication, provided it is legal and ethical to do so. Since the data are not freely available, please provide an appropriate contact (web or email address) for enquiries (please note that this cannot be a study author).

Title: Please revise your title according to PLOS Medicine's style. Please place the study design in the subtitle (ie, after a colon). We suggest “Evaluation of the Growth Assessment Protocol (GAP) for antenatal detection of small for gestational age: the DESiGN cluster randomised trial” or similar

Please remove the study summary (lines 56-71) and replace with an Author Summary (see below) to follow your Abstract. 

Abstract:

Please report your abstract according to CONSORT for abstracts, following the PLOS Medicine abstract structure (Background, Methods and Findings, Conclusions) http://www.consort-statement.org/extensions?ContentWidgetId=562

Abstract Background: Please provide the context of why the study is important. The final sentence should clearly state the study question.

Abstract Methods and Findings:

Please provide brief demographic details of the study population (e.g. age, ethnicity, etc)

Please be more specific regarding dates of the study for baseline data collection and randomisation. Please include a summary of adverse events if these were assessed in the study.

In the last sentence of the Abstract Methods and Findings section, please describe 2-3 of the main limitations of the study's methodology."

Abstract Conclusions:

Please begin your Abstract Conclusions with "In this study, we observed ..." or similar, to summarize the main findings from your study, without overstating your conclusions. Please emphasize what is new and address the implications of your study, being careful to avoid assertions of primacy. 

Author Summary:

In the final bullet point of ‘What Do These Findings Mean?’, please describe the main limitations of the study in non-technical language.

Please rename ‘Background’ to ‘Introduction’ 

Methods:

Please refer to your prospective protocol / analysis plan early in the Methods section. Legends for these files should be included at the end of your manuscript. Changes in the analysis-- including those made in response to peer review comments-- should be identified as such in the Methods section of the paper, with rationale. If a reported analysis was performed based on an interesting but unanticipated pattern in the data, please be clear that the analysis was data-driven.

Please include your Ethics statement within the Methods section.

Please ensure that the study is reported according to the CONSORT guideline, and include the completed CONSORT checklist as Supporting Information. Please add the following statement, or similar, to the Methods: "This study is reported as per the Consolidated Standards of Reporting Trials (CONSORT) statement (S1 Checklist)." The CONSORT guideline can be found here: http://www.consort-statement.org/ When completing the checklist, please use section and paragraph numbers, rather than page numbers which will likely no longer correspond to the appropriate sections after copy-editing.

Line 223 - please clarify what is meant by ' The utilisation of ultrasound has also been described’

Results: 

Please provide the actual numbers of events for the outcomes (this may be in the Tables, and not necessarily each time they're mentioned).

Please be very clear in the main text which results correspond to which arm. 

Discussion:

Please present and organize the Discussion as follows: a short, clear summary of the article's findings; what the study adds to existing research and where and why the results may differ from previous research; strengths and limitations of the study; implications and next steps for research, clinical practice, and/or public policy; one-paragraph conclusion.

Please remove all subheadings within your Discussion e.g. Main findings

Lines 380, 411, 462 - please temper assertions of primacy by adding ‘to the best of our knowledge’ or similar 

Figures:

Please provide titles and legends for all figures (including those in Supporting Information files).

Tables (including those in Supporting Information files):

Please relabel Box 1 as Table 1 

Please present numerators and denominators for percentages.

When a p value is given, please specify the statistical test used to determine it in the table legend.

Please define all abbreviations used in the table legend of each table.

References:

Please ensure that journal name abbreviations match those found in the National Center for Biotechnology Information (NCBI) databases, and are appropriately formatted and capitalised.

Please also see https://journals.plos.org/plosmedicine/s/submission-guidelines#loc-references for further details on reference formatting. 

Where website addresses are cited, please specify the date of access. 

Supplementary files: 

Please provide titles and legends for each individual table and figure in the Supporting Information.

Please see https://journals.plos.org/plosmedicine/s/supporting-information for our supporting information guidelines. 

Comments from the reviewers:

Reviewer #1: Thank you for allowing me to review the submitted manuscript. The authors are to be commended on try to prospectively study such a complex intervention at scale. There are however a few areas of concern that I think need to be addressed.

Methodololgy

-Despite reading reading this section several times I'm still nor entirely certain how the antenatal detection rate was devised. Was this from retrospective plotting of utlrasound EFWs onto chart or from a sample of notes. For readers to form opinions on the study then this section needs to be very clear and unambigous which it is not in my view currently.

-Standard care. There is no attempt to define what GAP is being compared and it is therefore difficult to judge whether GAP had a realistic chance of success (see below) following intervention. In my view the authors should provide more detail on what standard care is and whether this was within certain parameters or very variable. It is interesting to note that the standard care arm seemed to changes more significantly in terms of scan frequency than the GAP arm. There is also no attempt to describe whether units were secondary or tertiary or involved in other measures to improve SGA detection.

Results

The primary outcome table 3 is not very easy to read or interpret in its current format and I would suggest reviewing the row legends. 

Is there any capacity to perform a sensitivity analysis using scans performed within 3 weeks of birth rather than performed throughout the 3rd trimester.

Discussion

Whilst the populations are similar there are some signficant differences and these are not discussed. From my reading of table 1 it would appear that the sites where the intervention occurred had more pakistani women and this might potentially allter the sensitivity of grow/standard care.

Table 2 demonstrates some surprising findings that are difficult to understand/believe for example the number of women not receiving anatomy scans between 18-24 weeks seems very low. There is also a huge increase in >24 weeks scan frequency in the standard care arm which is not even mentioned. This increase means that 77% of women in standard care have a scan though there is no description of when during the third trimester these scans occur. The big problem here in my mind is that the primary function of GAP is not managing women who are having growth scans, but improving the detection of SGA in low risk women, by more accurately targetting scans. However when scan frequency is so high it seems unlikely any package of measures will have an effect on SGA detection rate. The authors should discuss this as it is a mojor limitation of the primary outcome. 

With such high % of scan usuage it is surprising that the detection rate was so low which suggest lots of process problems as briefly discussed in terms of historical scan timing. It is however difficult to judge the effecrtiveness of GAP if the recommended scan frequency has not been implemented. Of interest there does not seem to be a big difference between customised and non-customised outcomes in any metric. This is potentially the most interesting finding as although I'm not convinced the authors have adequately tested GAP, they have demonstrated that customisation does not really make a difference to any outcome. This should be highlighted and discussed in my view.

I note the stillbirth seems lower in the intervention arm, but no mention is made in the discussion about the pre-implementation phrase during which the standard care group had higher levels.

Summary

This study has potential practice chaging findings and I am very keen to see it published, but currently there are some significant changes requred in my view prior to publication.

Reviewer #2: Statistical review

This paper reports a cluster randomised trial investigating implementation of the Growth Assessment Protocol on detection of small for gestational age during pregnancy.

The trial was reported well and I had only minor comments.

1. Abstract "(instead of patients or subjects)" - presume this was leftover text?

2. Abstract - providing a brief summary of the low observed false positive detections of SGA might be of interest to aid interpretation of the results.

3. Page 14 line 314 'with obesity (15.7% vs 18.1%)' I initially read this as restricted to women of white ethnicity - a colon after 'less women' might help avoid misreads.

4. Page 15 line 343 - I feel it should be pointed out here that two marginally significant secondary outcomes out of 26 is consistent with what would be expected by chance if there were no benefit of intervention for any of the outcomes (to be fair the authors do point out this in the methods).

5. Discussion: Table 2 seems to show huge improvements in control clusters during the outcome period, compared to the baseline period. This might be worth reflecting on in the discussion. It wasn't clear to me why there were generally big differences between control and intervention clusters in the pre-randomisation period though?

James Wason

Reviewer #3: This is a well designed and executed study assessing the utility of the GAP program is the detection of fetal growth impairment (Defined as SGA, <10thcentile). The study was complicated by the roll-out of the Safer Baby Bundle, albeit the participating hospitals were exempted. The authors acknowledge that there would have been some contamination from that roll-out. 

The methodology is well described and appropriate. The results are appropriately reported.

I would make the following comments:

1. The standard care hospitals appeared to have a significant change in key outcomes from the pre-randomisation phase to the outcome period (ie % pregnancies with 18-24 week scan; % pregnancies with a scan >24 weeks; % pregnancies with a scan >24 weeks +EFW; % pregnancies with no scan). these changes suggest changes in care provision between these two times, in the direction of improved care. The authors don't discuss this effect. Could this not explain why there is no difference in outcome between the two groups of hospitals? A discussion of this would be useful.

2. The authors report tht the rate of stillbirth and perinatal mortality were reduced in the intervention group of hospitals but not the standard care group. What they don't comment on is that the rates fell to those of the standard care hospitals. Nonetheless, this is an important finding. As the authors comment, the whole point of detecting SGA is to reduce stillbirth. I think this finding should be in the Abstract and in the Main Findings section with further discussion in the Implications.

3. The intervention led to earlier birth than standard care. This is a common finding among studies exploring better detection of SGA. Indeed, in this study the false positive rate increased in both groups too. Earlier delivery of SGA, but not false positive AGA, effect has potential to cause longer term cognitive harm. This was recently reported in JAMA (Selvaratnam 2021;326(2):145-153). A comment on the possibility of harm - not just lack of utllity - would be important.

Reviewer #4: This cluster randomized trial evaluated the effectiveness of implementing the GAP program for antenatal detection of SGA births in maternity care hospitals in the UK. The research question is a highly important one, as this program appears to have been widely adopted into routine care across the UK without high-quality evidence of its effectiveness, and findings from this study suggest that the program may not in fact be achieving its intended objective. The trial was a pragmatic one, evaluating the effectiveness of the program as implemented in a real-world setting (as opposed to evaluating the efficacy of the program if implemented under optimal conditions), which is most relevant to informing roll-out of the program nationally and internationally.

The study findings are a challenge to interpret because of several issues arising during the trial, most notably, the concurrent national introduction of the Saving Babies Lives care bundle. The adoption of this care bundle means that the type of care provided to the standard of care comparison group likely changed across the study period, making it unclear to what clinical care settings the adverse outcome rates in the comparison group are generalizable to, and decreasing the value of the information on baseline rates. I do sympathize with the investigators, as the unfortunate timing of introduction of this care bundle was beyond their control, and as a new trial to address this question now seems unlikely, believe the trial nevertheless provides the best available information to help assess the merits of this program. 

Main comments

1) My first concern relates to the two stage cluster-summary statistical analysis approach used, primarily as it relates to the less common secondary outcomes. If I understand correctly (without remote access to the textbook cited in the Supporting Information), the first stage of the two-stage cluster analysis calculates a summary value for each site (i.e., a site-specific mean rate (percent) of each outcome), adjusted for the listed covariates, and these rates (percents) are then put into a linear regression model estimating the difference in rate (percent) between treatment and control groups in stage 2. My concern is that for uncommon secondary outcomes, the site-specific rates estimated in step 1 will be very unstable because of low numbers of events at each site. For example, with 11,096 births in the intervention arm, and a stillbirth rate of 3.1 per 1000, there will only be ~34 stillbirths across all 5 intervention sites, meaning that the site-specific rates are being estimated from only 6-7 events per site. One stillbirth more, or less, per site will have an outsized impact on a site's rate, which in turn, will have a major impact on the comparison of rates between sites given the small number of sites. 

I am not a trial statistician, but it would seem to me that for rare outcomes, an approach should be chosen that does not depend on accurate estimation of the hospital-specific rates. I agree that a multi-level model might be challenging given the small number of clusters, but is it not possible to build a single model using the original individual-level data that controls for clustering by hospital using GEE (or even an indicator variable for hospital), and uses marginal estimation to calculate risk differences between treatment arms (e.g., as per the approaches compared here: https://bmcmedresmethodol.biomedcentral.com/articles/10.1186/s12874-016-0217-0). Please better justify the choice of analytic approach as it relates to rare outcomes.

2) On a related note, please expand the description of the analytic approach in the Supporting Information to clarify how the two-step approach accounts for differences in cluster sizes. That is, the site-specific rate from a large site will be more precise than that from a small site; were the summary values in the second step weighted to account for the number of births at that site?

3) The amount of missing data in this trial is concerning. Although I agree that using existing hospital databases can be a reasonable strategy for efficiently collecting participant and outcome data for a large number of trial participants, the amount of missing data in this trial was much higher than I typical expect from clinical perinatal databases (e.g., 37% missing values for a key pregnancy complication such as pre-eclampsia during the outcome period in the standard of care group), and for variables that should not have missing data (e.g., vital status at discharge [neonatal mortality, stillbirth], based on the lack of 'n' provided in Table 4). This raises concerns for me about data quality and the appropriateness of multiple imputation given high rates of missingness. What efforts were made to reduce missing data and ensure data quality? Were chart audits considered to eliminate missing data for important health outcomes such as stillbirth and neonatal mortality? 

4) The Supporting Information section on data management states, "For one cluster, ultrasound measurement data were not available for the baseline period. The proportion of SGA infants detected antenatally (by both definitions) at baseline for this cluster was imputed based on a model fitted to data from the other clusters predicting the number of infants detected based on the number of pregnancies with an ultrasound scan after 24 completed weeks". Imputing values for an entire site with missing data does not seem like a good solution given the between-site variation observed in the study. Did the study team consider conducting a chart review for a random sample of deliveries from this site to at least have some real data from the site to include in the imputation model? Further, more information on this lack of a key outcome variable seems important given the adjustment for baseline values in the final model: a) was this cluster in the treatment or control group, and b) were sensitivity analyses done excluding this cluster from analyses?

5) While appreciating that an economic analysis is likely beyond the scope of this manuscript, it does seem important to include some mention of the costs of this intervention. If a program is free, shows no evidence of causing harm, and there is a suggestion that it might improve an important health outcome like perinatal mortality, one might argue that it is not unreasonable to still implement the program despite the lack of evidence for its effectiveness, given the potential for benefit. However, if the program uses a meaningful amount of resources, then implementation of this program is much less justifiable, given that it would be detracting resources from other proven interventions. Can the authors comment in general terms on the costs of GAP implementation for those readers less familiar with the program? If implemented outside of the trial setting, does the Perinatal Institute provide its services for free, or are hospitals required to pay for the Institute's support? How much staff time was required/recommended for training and implementation? For ongoing running of the program?

6) The trial findings were not entirely unexpected, given previous literature demonstrating that customised growth charts (the core element of the GAP protocol) are not meaningfully better at identifying high-risk infants than non-customised ultrasound fetal growth charts (customised charts generally only appear to be better at identifying high risk infants when compared to birthweight charts [i.e., charts derived from the weights of live births rather than estimated fetal weights from ongoing pregnancies], which do not reflect the charts used antenatally to monitor fetal growth and are biased at preterm ages because of the correlation between preterm birth and fetal growth restriction). Citing some of this literature in the Discussion, particularly those studies comparing ultrasound estimated fetal weights classified using customised and non-customised standards, seems relevant to help interpret the plausibility of trial findings. E.g., Am J Obstet Gynecol 2018:218:S738-S744; Am J Epidemiol 2011;173:539-543.

Minor comments

1) It would be helpful to include a diagram of the study timeline in the Supplementary Material (pre-intervention, implementation, washout, and outcome collection periods by calendar time at each of the sites) - the description in the main body of the text was somewhat hard to follow, and given the concurrent roll-out of the Saving Babies Lives care bundle, it would be helpful to know the calendar timing of outcome collection at trial sites in relation to this bundle.

2) Please confirm that the study outcomes were included in the imputation models in the description of multiple imputation in the Supporting Information.

3) Presentation of results: an "adjusted difference 2.2%, 95% CI -6.4% to 10.7%" could be interpreted by some to be a relative change (i.e., an increase of 2.2%) rather than an absolute change in percentage points (which I believe is the correct interpretation based on a linear regression model). If so, the authors may wish to rephrase this as "2.2 more SGA infants detected per 100 births" for clarity for this and all outcomes.

4) The stillbirths are of considerable interest, and although the authors were appropriately cautious in interpreting the results for this outcome given the number of secondary analyses conducted, I believe that readers may be tempted to focus on this outcome. Including more information on these cases would be helpful, such as presenting the counts in each group (and counts of missing data), as well as a summary of cause of deaths.

5) The references in the Supporting Information appear to be incomplete; please update instances of "ref Mx" and "ref data Mx" (page 3) with the appropriate citation.

[LINK]

---

## [Decision Letter · Decision Letter 2]

8 Jan 2022

Dear Dr. Vieira,

Thank you very much for submitting your manuscript "Evaluation of the Growth Assessment Protocol (GAP) for antenatal detection of small for gestational age: the DESiGN cluster randomised trial." (PMEDICINE-D-21-03941R2) for consideration at PLOS Medicine. 

Your paper was re-reviewed by three reviewers, including the statistical reviewer, and discussed among the editorial team and with an academic editor with relevant expertise. The reviews are appended at the bottom of this email and any accompanying reviewer attachments can be seen via the link below:

[LINK]

The reviewers have identified unresolved issues relating to the statistical analysis and a lack of detail regarding what constituted standard care. As such, we are unable to accept the manuscript for publication in its current form, but we would like to consider a revised version that fully addresses the reviewers' comments. Obviously we cannot make any decision about publication until we have seen the revised manuscript and your response, and we plan to seek re-review by one or more of the reviewers. 

In revising the manuscript for further consideration, your revisions should address the specific points made by each reviewer. Please also check the guidelines for revised papers at http://journals.plos.org/plosmedicine/s/revising-your-manuscript for any that apply to your paper. In your rebuttal letter you should indicate your response to the reviewers' and editors' comments, the changes you have made in the manuscript, and include either an excerpt of the revised text or the location (eg: page and line number) where each change can be found. Please submit a clean version of the paper as the main article file; a version with changes marked should be uploaded as a marked up manuscript.

We expect to receive your revised manuscript by Jan 31 2022 11:59PM. Please email us (plosmedicine@plos.org) if you have any questions or concerns.

We look forward to receiving your revised manuscript. 

Sincerely,

Louise Gaynor-Brook, MBBS PhD

PLOS Medicine

plosmedicine.org

Comments from the reviewers:

Reviewer #1: Thank you for the detailed responses and changes made to the manuscript in response to mine and others reviews.

My remaining area of concern regards the information on standard care provided in the mansuscript in line 215-229 and in the response to review. 

The authors state that-

"This is a pragmatic based trial on 'standard care'; we believe attempting to 'standardise' standard care in this study would create an artificial comparison and potential for change of practice given the study processes (contamination). Even though we agree it is challenging to ascertain based on our trial to what is an 'optimal' standard practice, we do believe practice in standard care clusters reflected national practice in maternity units not implementing GAP, including national variation."

-and then go on to state that a detailed review of the standard care local guidelines is already available in a prior publication (not referenced). Having looked for this I cannot easily find it to assess whether the reader can determine the range of standard care applied, but still think more information needs to be provided in this manuscript. Given there are only a small number of units in the study standard arm it would not require much additional work to include for example how many units employed uterine artery Dopplers or not, whether surveillance scan frequency was 3, 4 weekly or longer or prespecified at certain gestations, and gestation of last planned surveillance scan. This information could all be included in a simple supplementary table. Due to the international reach of PLOS med and the fact that this study's findings will be of interest to a range of healthcare systems outside of the UK I do not think quoting the RCOG guidelines and potential adherence to this is sufficient and still requires further work.

Reviewer #2: Thank you to the authors for addressing my previous comments well. I have no further issues to raise.

Reviewer #4: I appreciate the authors' thoughtful and detailed responses, which have satisfactorily answered most of my initial concerns.

I do, however, remain concerned about the use of the two-stage approach in the context of rare events (i.e, estimation of site-specific rates based on very low counts of events). The authors describe the two-stage approach in their Statistical Analysis (supporting information) as: "Firstly, the cluster summary values were adjusted for the ethnicity, age and parity of the individual participants (these are residuals from comparing the observed summary values with those predicted from a model fitted to all participants)... In the second stage, linear regression analysis (ANCOVA) was undertaken in which the adjusted cluster-summary values for an outcome in the trial outcome period were compared between intervention and standard care arms..." It is thus unclear to me how the response to reviewer comments can then claim "However, we disagree that the method requires accurate estimation of hospital-specific rates"-- unless I am misunderstanding, the adjusted hospital-specific rates are precisely what is being compared between groups, so an inaccurate or unstable rate would be expected to influence results. Especially if there is uncertainty in the literature as to which is the best approach, it would be helpful to include results showing the results when using a one-stage approach as supporting information. Confirmation that findings are robust to the type of analytic method would greatly strengthen confidence in the study conclusions.

[LINK]

---

## [Decision Letter · Decision Letter 3]

14 Apr 2022

Dear Dr. Vieira,

Thank you very much for re-submitting your manuscript "Evaluation of the Growth Assessment Protocol (GAP) for antenatal detection of small for gestational age: the DESiGN cluster randomised trial." (PMEDICINE-D-21-03941R3) for review by PLOS Medicine.

I have discussed the paper with my colleagues and the academic editor and it was also seen again by three reviewers. I am pleased to say that provided the remaining editorial and production issues are dealt with we are planning to accept the paper for publication in the journal.

[LINK]

We look forward to receiving the revised manuscript by Apr 21 2022 11:59PM.   

Sincerely,

Louise Gaynor-Brook, MBBS PhD

PLOS Medicine

plosmedicine.org

Requests from Editors:

General comments:

To help us extend the reach of your research, please provide any Twitter handle(s) that would be appropriate to tag, including your own, your coauthors’, your institution, funder, or lab.

Data availability:

The Data Availability Statement (DAS) requires revision. Please remove “request made to the Chief Investigator (Prof Dharmintra Pasupathy) as, in the interests of transparency and reproducibility, a study author cannot be a contact person for the data or be responsible for approving access to data. 

Abstract:

Abstract Background: please temper assertions of primacy by adding ‘to the best of our knowledge’ or similar with relation to ‘there are no reported randomised control trials.’

Abstract Methods and Findings:

Please clarify whether this should be November 2015 or 2016 (noting discrepancy between the Abstract and Methods section)

Please add a sentence to summarise the secondary outcomes 

Please revise to <10th centile 

Please define CI at first use

Author Summary:

Please temper assertions of primacy by adding ‘to the best of our knowledge’ (or similar) with relation to ‘This first randomised control trial’

In the final bullet point of ‘What Do These Findings Mean?’, please describe the main limitations of the study in non-technical language.

Methods

Thank you for providing a CONSORT checklist. When completing the checklist, please use section and paragraph numbers, rather than page numbers which will likely no longer correspond to the appropriate sections after copy-editing.

Results: 

Line 367 (and throughout) - please revise to ‘there were fewer women..’, ‘fewer stillbirths’, etc 

Where adjusted analyses are presented, please specify which factors are adjusted for, and provide the unadjusted analyses.

Discussion:

Line 443 - please temper assertions of primacy by adding ‘to the best of our knowledge’ or similar 

Tables:

Tables 2, 3, 4, 5, S3A, S3B, S3C, S4A, S4B - When a p value is given, please specify in the table legend the statistical test used to determine it.

Tables 3, 4, 5, S1C, S3A, S3C, S4B - Please present numerators and denominators for percentages

Supplementary Table 2 appears to be missing

Comments from Reviewers:

Reviewer #1: Thanks for including the information requested which as suspected has revealed that there is no single "standard care" in the different sites not randomised to GAP. This likely means that GAP has been compared to multiple different preexisting detection rates. However, I think as this information is now clear for the reader there is no reason for the manuscript to go forward for publication in my view and look forward to the interest and debate it will create.

Reviewer #2: I presume I've been asked to re-review (after commenting that I was happy with the R2 version) is to comment on the two-stage vs one-stage approach. I'm afraid I am not knowledgable about that: I asked a colleague who works on cluster randomised trials and they are not aware of any investigation that has shown an issue with two-stage approaches. They did not feel there was a reason to favour using a one-stage approach instead of a two-stage.

I think the authors having added an additional analysis and discussion has addressed this issue to my satisfaction, but the other reviewer perhaps knows more about the issue than I do.

Reviewer #4: I have no further suggestions for the manuscript; this revised version of the manuscript addresses my previous concerns.

[LINK]

---

## [Editor Report · Decision Letter 4]

29 Apr 2022

Dear Dr Vieira, 

On behalf of my colleagues and the Academic Editor, Prof. Jenny Myers, I am pleased to inform you that we have agreed to publish your manuscript "Evaluation of the Growth Assessment Protocol (GAP) for antenatal detection of small for gestational age: the DESiGN cluster randomised trial." (PMEDICINE-D-21-03941R4) in PLOS Medicine.

PRESS

Sincerely, 

Louise Gaynor-Brook, MBBS PhD 

PLOS Medicine